# Coordination environments of Pt single-atom catalysts from NMR signatures

Jonas Koppe[1], Alexander V. Yakimov[2,3], Domenico Gioffrè[2], Marc-Eduard Usteri[2,3], Thomas Vosegaard[4], Guido Pintacuda[1], Anne Lesage[1], Andrew J. Pell[1], Sharon Mitchell[2,3], Javier Pérez-Ramírez[2,3 ✉] & Christophe Copéret[2,3 ✉]

Supported metal catalysts that integrate atomically dispersed species with controlled structures lie at the forefront of catalytic materials design, offering exceptional control over reactivity and high metal utilization, approaching the precision of molecular systems[1–3]. However, accurately resolving the local metal coordination environments remains challenging, hindering the advancement of structure–activity relationships needed to optimize their design for diverse applications[1,2]. Although electron microscopy reveals atomic dispersion, conventional spectroscopic methods used in heterogeneous catalysis only provide average structural information. Here we demonstrate that $^{195}$Pt solid-state nuclear magnetic resonance (NMR) spectroscopy is a powerful tool for characterizing atomically dispersed Pt sites on various supports, so called single-atom catalysts (SACs). Monte Carlo simulations allow the conversion of NMR spectra into SAC signatures that describe coordination environments with molecular precision, enabling quantitative assessment of Pt-site distribution and homogeneity. This methodology can track the influence of synthetic parameters, uncovering the impact of specific steps and support types, and can also monitor changes upon reaction. It offers critical insights for the reproducible development of SACs with targeted structures. Beyond SACs, this approach lays the foundation for studying more complex architectures, such as dual-atom or single-cluster catalysts, containing various NMR-active metals.

Catalytic materials featuring atomically dispersed metal species are central to advancing sustainable chemical processes, offering exceptional precision in reactivity. In particular, single-atom catalysts (SACs) based on platinum (Pt) are known to catalyze a range of transformations across electro-, photo- and thermo-driven processes[3–5]. Among various supports, functionalized carbons, for example, N-doped carbon (NC) are widely applied due their availability and tunability. These porous solids stabilize atomically dispersed Pt by means of anchoring to the support, presumably through N sites, directly paralleling the realm of molecular coordination complexes[6]. Although the catalytic properties have been linked to differences in binding sites across supports[7,8], it remains challenging to: (1) assess specific coordination environments for structure–activity relationships; (2) evaluate metal–site homogeneity, evolution during synthesis and variation as a function of NC supports; (3) ensure synthetic protocol reproducibility[9]; and (4) monitor the evolution of catalyst structures upon reaction.

The lack of molecular-level information for SACs contrasts with their homogeneous or even enzymatic counterparts, hindering implementation of rational design strategies. Molecular catalysts can be fully characterized by combining elemental analysis, spectroscopic techniques and, ultimately, single-crystal X-ray diffraction. However, similar information remains scarce for SACs: high-angle annular dark field scanning transmission electron microscopy (HAADF-STEM) can helpascertain metal dispersion given suitable contrast, but X-ray photoelectron spectroscopy (XPS) and X-ray absorption spectroscopy (XAS) provide only partial or average information regarding metal–site structure[10], which is often inferred by intuition and/or density-functional theory (DFT) computational modelling[5,11].

Notably, nuclear magnetic resonance (NMR) spectroscopy, an element-selective technique, should be ideally suited to probe the local structure of isolated metal centres in SACs, particularly for $^{195}$Pt, which is a spin-1/2 nucleus with a large chemical shift range, sensitive to the local environment of Pt. In solution, the signal is a sharp line with a resonance frequency referred to as the isotropic chemical shift ($\delta_{iso}$), which varies across a large frequency range, from positive values for Pt(IV), to $-1{,}500 \geq \delta_{iso} \geq -4{,}500$ ppm for Pt(II) (ref. 12) and $\delta_{iso} \leq -4{,}000$ ppm for Pt(0). In the solid state, NMR spectra are more complex but provide unique information about the electronic structure of metal sites[12–18]. The NMR spectrum is associated with a characteristic lineshape, referred to as a powder pattern (thereafter called pattern, for brevity), which can be described by a chemical shift (CS) tensor, characterized by its three principal-axis components, $\delta_{11} > \delta_{22} > \delta_{33}$, whose average corresponds to $\delta_{iso} = 1/3(\delta_{11} + \delta_{22} + \delta_{33})$. The overall pattern linewidth is described by the span $\Omega = \delta_{11} - \delta_{33}$,

[1]Centre de RMN à Très Hauts Champs de Lyon, CNRS/Ecole Normale Supérieure de Lyon/Université Claude Bernard Lyon 1, Villeurbanne, France. [2]Department of Chemistry and Applied Biosciences, ETH Zürich, Zürich, Switzerland. [3]NCCR Catalysis, Zürich, Switzerland. [4]Department of Chemistry and Interdisciplinary Nanoscience Center, Aarhus University, Aarhus C, Denmark. ✉e-mail: jpr@chem.ethz.ch; ccoperet@ethz.ch

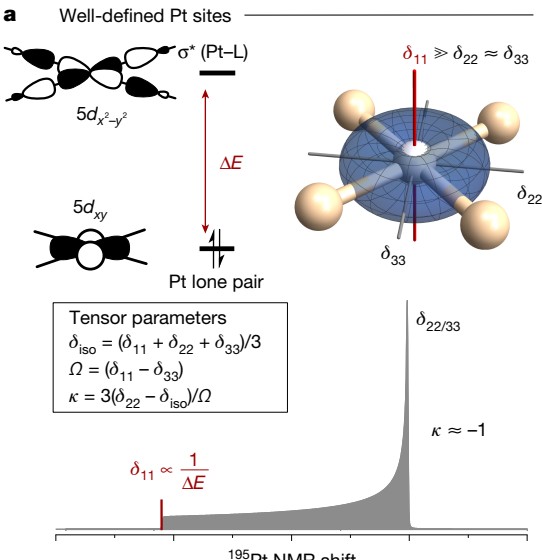

**a** Well-defined Pt sites

**b** Pt-site distributions

**Fig. 1 | $^{195}$Pt NMR of square-planar Pt(II) sites. a**, For square-planar Pt(II) complexes, frontier molecular orbitals, for example, filled lone pair and empty $\sigma^*$ (Pt–L) orbitals, translate into a $^{195}$Pt CS tensor with a dominant contribution of the $\delta_{11}$ component perpendicular to the plane, that is, $\delta_{11} \gg \delta_{22} \approx \delta_{33}$, leading to large overall linewidths (span $\Omega = \delta_{11} - \delta_{33}$), and oblate tensor shapes (skew $\kappa \approx -1$). The magnitude of $\delta_{11}$ drives the isotropic chemical shift (centre of gravity) $\delta_{iso} = 1/3(\delta_{11} + \delta_{22} + \delta_{33})$ and is modulated by the energy difference $\Delta E$ between the Pt lone pair and the $\sigma^*$ (Pt–L) orbital, which depends on the nature of the ligand bound to Pt. A smaller $\Delta E$, associated with a more electronegative ligand, results

in a larger $\delta_{11}$ and $\Omega$. **b**, The heterogeneity of Pt(II) sites in supported catalysts is expected to cause a distribution of $^{195}$Pt CS tensors, yielding a less well-defined, 'blurred' $^{195}$Pt NMR signature, which can be described by average CS-tensor parameters $<\delta_{iso}>$, $<\Omega>$ and $<\kappa>$. In addition, the variation of the $\delta_{11}$ component, and with that the variation of $\Delta E$, can be described by the chemical-environment heterogeneity $\sigma$. The relative variation of $\delta_{11}$ compared with $\delta_{22}/\delta_{33}$ can be captured by $\rho$, representing the heterogeneity of the local geometry. In structural diagrams, Pt atoms sit at the centre and are grey.

which can vary from 0 ppm for an isotropic Pt coordination, for example, tetrahedral Pt(0) or octahedral Pt(IV) complexes, to more than 10,000 ppm for square-planar Pt(II), depending on specific ligand environments[12–16]. Furthermore, the skew, $\kappa = 3(\delta_{22} - \delta_{iso})/\Omega$, describing the shape of the powder pattern[19], contains information on the Pt-site symmetry with the minimum value $\kappa = -1$, reflecting the oblate square-planar Pt(II) structural motif, and the maximum $\kappa = +1$ observed for prolate tensor shapes (Fig. 1 and caption). Overall, $^{195}$Pt NMR patterns, defined by $(\delta_{iso}/\Omega/\kappa)$, are precise reporters of Pt local environment and electronic structure (configuration/geometry/ligands). Any changes in ligands or local geometries will affect these parameters, tracing the evolution of coordination environment and speciation.

Despite this potential, $^{195}$Pt NMR often produces extremely broad signals with a low signal-to-noise ratio, which is worsened by the typically low Pt content in SACs (typically below 5 wt%) and the inherently low sensitivity of solid-state NMR. Furthermore, the width of $^{195}$Pt NMR patterns easily exceeds frequency ranges of conventional radio-frequency pulses, requiring powder patterns to be recorded stepwise[20] in several experiments. Consequently, experimental times extending to one month or more[21] have prevented the successful application of $^{195}$Pt NMR to SACs.

Here we demonstrate how state-of-the-art ultra-wideline NMR methodology[22] under static[23–25] and magic-angle spinning[26] (MAS) conditions at low temperatures[27–30] and fast repetition rates[31] (Extended Data Fig. 1) enables the acquisition of complete $^{195}$Pt NMR spectra in reasonable times (hours to a few days) for SACs with Pt contents down to about 1 wt%. This approach establishes $^{195}$Pt solid-state NMR spectroscopy as the method of choice for characterizing the Pt atomic environments. DFT calculations combined with Monte Carlo simulations show that $^{195}$Pt NMR patterns provide information about coordination environments and site distributions. This methodology allows the quantitative monitoring of coordination environment evolution upon synthesis and catalysis, and highlights supporting effects, providing a guideline for

understanding differences between catalytic materials and synthetic protocols, and possible origins of deactivation.

Here atomically dispersed platinum on nitrogen-doped carbons (Pt@NC) were chosen as prototypical SACs. Evaluating the sensitivity of our $^{195}$Pt NMR approach towards subtle changes in coordination environments was possible thanks to the synthesis of SACs with 15 wt% as well as 5 wt% of Pt (ref. 32), denoted Pt@NC-15 and Pt@NC-5. We studied both samples after a first (at 200 °C) and a second (at 550 °C) annealing step, yielding an initial library of four materials with different Cl/Pt ratios in the range of 4 to 1 (Extended Data Table 1). HAADF-STEM images (Extended Data Figs. 2–4) confirm that Pt@NC samples contain isolated Pt atoms, with an estimated surface atom density of 0.35 atoms nm$^{-2}$. Furthermore, XPS suggests that Pt is predominantly found in +II oxidation state with pyridinic nitrogen in its coordination environment (Extended Data Fig. 5). Here we demonstrate how $^{195}$Pt solid-state NMR spectroscopy enables derivation of an atomic-scale picture of the Pt sites (coordination environment, electronic structure, site homogeneity) and can track changes in these characteristics across synthetic protocols, which is a first step towards quality control for precision catalysis.

## $^{195}$Pt NMR of molecular and supported Pt sites

The static and MAS $^{195}$Pt-ultra-wideline NMR methodologies were first employed to measure the NMR patterns of four molecules with local Pt environments close to those expected in Pt@NC: $K_2PtCl_4$, $cis$-Ptpy$_2$Cl$_2$, Pt(NH$_3$)$_4$Cl$_2$ and $cis$-PtMe$_2$tmeda (Fig. 2a–d). Their $^{195}$Pt NMR patterns, that is, sharp spectral features under static conditions or a single manifold of narrow spinning sidebands under MAS conditions (20 kHz), testify the presence of a single, molecularly defined square-planar Pt(II) site associated with specific and characteristic tensor parameters $(\delta_{iso}/\Omega/\kappa)$: $\delta_{iso} = -1,650$ to $-3,840$ ppm, $\Omega = 10,420$ to 4,600 ppm, $\kappa \approx -1$ (oblate). The decrease of both $\delta_{iso}$ and $\Omega$ is mostly driven by a decrease of $\delta_{11}$, which lies perpendicular to the Pt–L plane

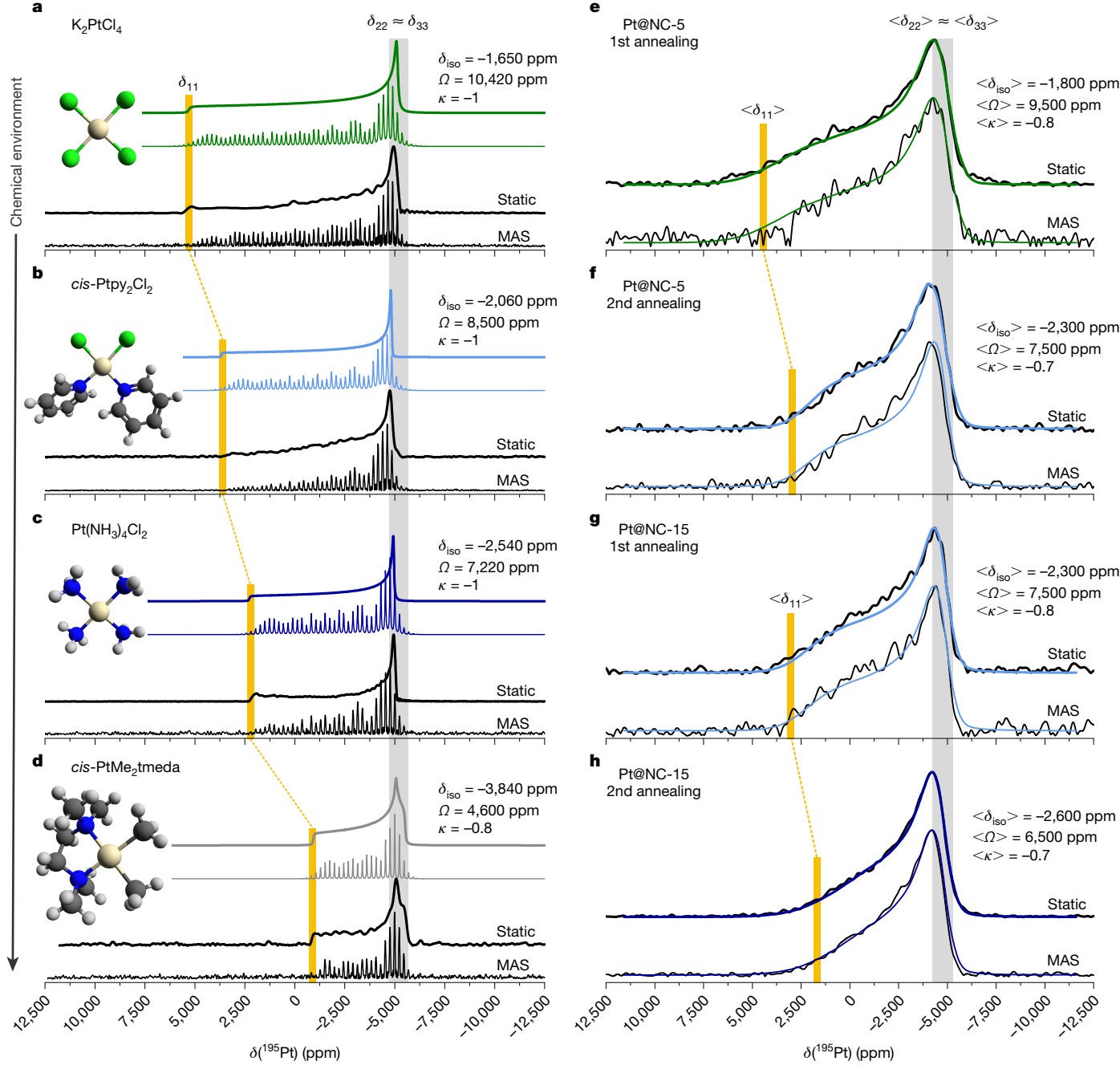

**Fig. 2 | Static and MAS $^{195}$Pt solid-state NMR characterization. a–d,** Molecular Pt references: potassium tetrachloroplatinate $K_2PtCl_4$ (**a**), *cis*-dichlorobis (pyridine)platinum *cis*-Ptpy$_2$Cl$_2$ (**b**), tetraamineplatinum chloride $Pt(NH_3)_4Cl_2$ (**c**) and *cis*-dimethylplatinum tetramethylethylenediamine PtMe$_2$tmeda (**d**) with the respective numerical model[35] (coloured lines) and CS-tensor parameters. **e–h,** Pt@NC samples with 5 wt% (**e,f**) and 15 wt% (**g,h**) Pt content after the first (**e,g**) and second (**f,h**) annealing step, respectively. The simulated lineshapes (coloured lines) are based on Monte Carlo simulations as described in the main

text. The average CS-tensor parameters are explicitly given. The three average CS-tensor components ($<\delta_{11}>$, $<\delta_{22}>$ and $<\delta_{33}>$) are likewise schematically indicated and correspond to the simulation input. Additional information regarding the experimental parameters, acquisition protocols, data processing, numerical modelling, the effect of different MAS frequencies and acquisition temperatures, as well as relaxation properties is provided in the Supplementary Information.

(Fig. 1), which, in turn, correlates with an increased number of N (and C) atoms bound to Pt upon substitution of chlorides by N-based or C-based ligands.

Next, we apply the ultra-wideline NMR methodologies to the Pt@NC samples. The static $^{195}$Pt NMR spectra (Fig. 2e–h) are reminiscent of the powder patterns observed for the Pt references, but with less well-defined spectral singularities, yet indicating square-planar Pt(II). The 'blurred' spectral features are indicative of heterogeneous local Pt environments, resulting in a distribution of CS-tensor parameters[33].

MAS (10 kHz) experiments do not lead to any notable changes in the appearance in any of these $^{195}$Pt NMR spectra (Fig. 2e–h), that is, no motional averaging of the anisotropic line broadening into a spinning sideband manifold is observed, contrary to the molecular references (Fig. 2a–d). This provides clear experimental evidence for a distribution of Pt sites and associated $\delta_{iso}$ (ref. 33), which is much broader than the MAS frequency. Even very fast MAS (50 kHz) does not lead to improved resolution, which suggests a $\delta_{iso}$ distribution exceeding 50 kHz ≈ 600 ppm (Extended Data Fig. 6).

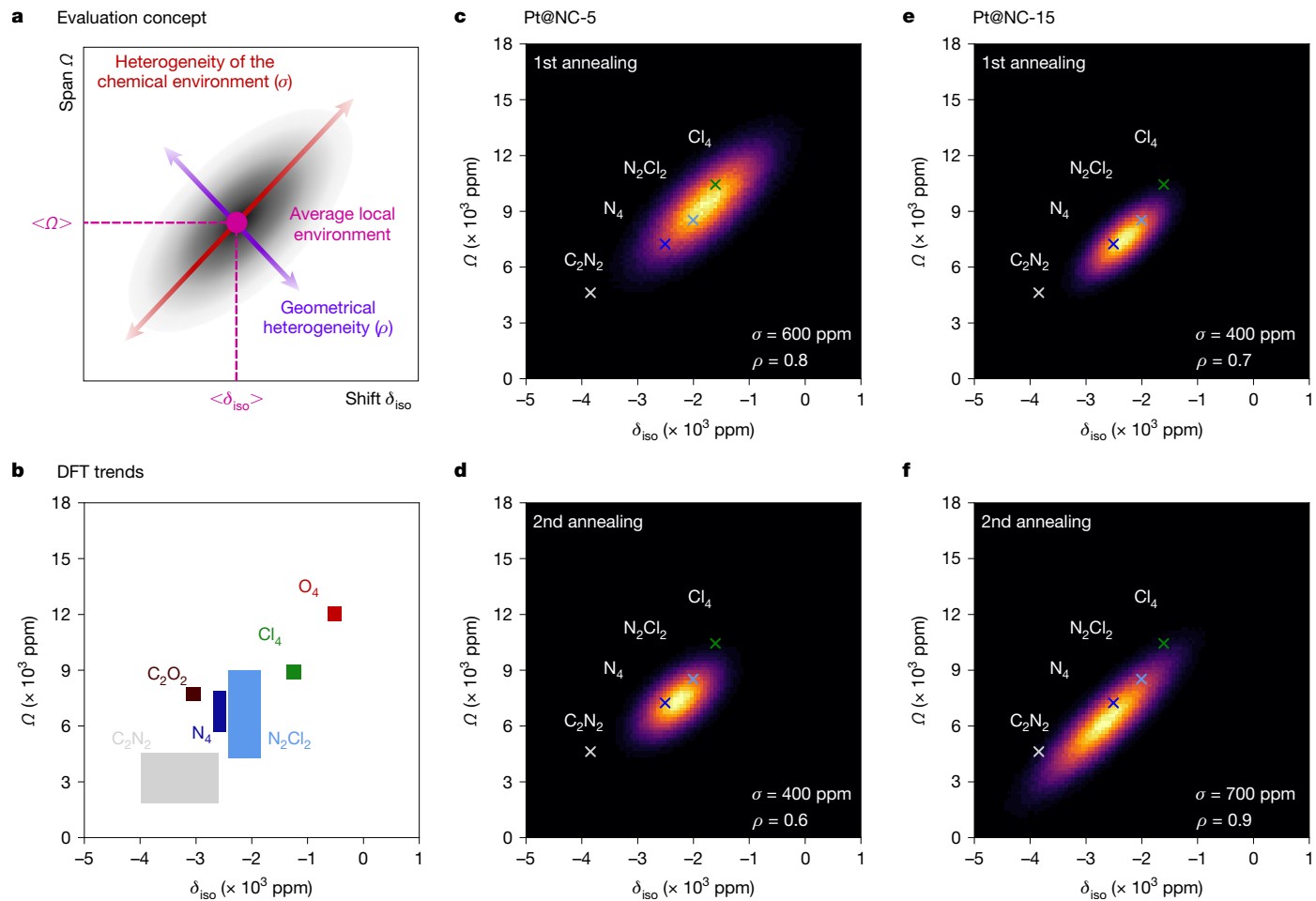

**Fig. 3 | Quality index of Pt coordination environments. a,** Scheme of the information provided by the $\delta_{iso}$–$\Omega$-space for Pt(II) environments. **b,** DFT trends for different chemical environments for square-planar Pt(II) sites (see Extended Data Fig. 7 for details). **c–f,** $\delta_{iso}$–$\Omega$-distribution maps from the Monte Carlo simulations performed to calculate the NMR lineshapes for Pt@NC-5 (**c,d**) and Pt@NC-15 (**e,f**) after the first and second annealing steps, respectively. The $Q_{ICE}$ parameters $\sigma$ and $\rho$ are respectively indicated. Crosses show the positions of the four molecular compounds from Fig. 2a–d K$_2$PtCl$_4$ (green), cis-Ptpy$_2$Cl$_2$ (light blue), Pt(NH$_3$)$_4$Cl$_2$ (dark blue) and cis-Pt(Me)$_2$tmeda (light grey).

## Quantitative indexation of Pt sites

Considering these broad lines, any further atomic-scale assessment of Pt sites across the Pt@NC samples requires a numerical model capable of describing the observed $^{195}$Pt NMR patterns, appropriately accounting for the non-uniform coordination environments. Here we consider a distribution of square-planar Pt(II) geometries with large variation of the longitudinal $\delta_{11}$ component, whereas both equatorial components ($\delta_{22}$ and $\delta_{33}$) vary much less[14]. This is accounted for by applying Gaussian distributions for the three principal-axis components, where the standard deviation for $\delta_{11}$ is different from that for $\delta_{22}$ and $\delta_{33}$. This results in five input parameters: the three expectation values for $<\delta_{11}>$, $<\delta_{22}>$ and $<\delta_{33}>$, and the two standard deviations $\sigma_{long}$ (for $\delta_{11}$) and $\sigma_{eq}$ (for $\delta_{22}$ and $\delta_{33}$). The corresponding distributions of CS tensors can be probed by Monte Carlo simulations, and the resulting spectra computed by summing all of the individual $^{195}$Pt NMR signals (see Supplementary Information section 5 for more details). For each Pt@NC sample, we report the average CS-tensor parameters ($<\delta_{iso}>$/$<\Omega>$/$<\kappa>$), representing the average Pt environment. Furthermore, we describe the heterogeneity of the chemical environment and local geometry of Pt by the respective $\delta_{iso}$–$\Omega$-distribution maps (Fig. 3): for chemical-environment heterogeneity, the variation of $\delta_{11}$ is dominant, and all individual CS tensors concentrate on a line with slope $\Delta\Omega/\Delta\delta_{iso} = 3$ (Fig. 3a). This is indeed observed for DFT-calculated and experimental data for square-planar Pt(II) (Fig. 3b and Extended Data Fig. 7), showing that $\delta_{iso}$ and $\Omega$ are

strongly correlated, as quantified by their linear correlation coefficient $\rho = 1$. The width of the distribution along the line, denoted by $\sigma$, therefore indicates the heterogeneity of the chemical environment.

Substantial $\delta_{22}/\delta_{33}$ variation is attributed to heterogeneity of the local Pt coordination geometry, as it does not result from the change of ligand environment, reducing the correlation between $\delta_{iso}$ and $\Omega$ (reduction of $\rho$). Both $\sigma$ and $\rho$ serve as a quantitative quality index for the Pt coordination environment ($Q_{ICE}$), describing heterogeneity in terms of chemical environment and local geometry, respectively. Together with the average CS-tensor parameters ($<\delta_{iso}>$/$<\Omega>$/$<\kappa>$), $Q_{ICE}$ describes a SAC signature, providing a precise description of the dominant local Pt structure and site distribution.

## Sensitivity to coordination environments

The lineshapes derived from the numerical model are in excellent agreement with the experimentally observed static and MAS $^{195}$Pt NMR patterns for Pt@NC (Fig. 2e–h). For all four samples, we indicate ($<\delta_{iso}>$/$<\Omega>$/$<\kappa>$) obtained from the Monte Carlo simulations; the corresponding $\delta_{iso}$–$\Omega$-distribution maps and the respective $Q_{ICE}$ ($\sigma$, $\rho$) are shown in Fig. 3c–f, where the discrete experimental data points for the molecular references with uniform environments are included.

The average skew values $<\kappa> = -0.7$–$0.8$ (close to $-1$), found across all Pt@NC samples, confirm the successful conversion of H$_2$Pt(IV)Cl$_6$ into square-planar Pt(II) sites during synthesis. Notably, $<\delta_{is}>$/$<\Omega>$, which

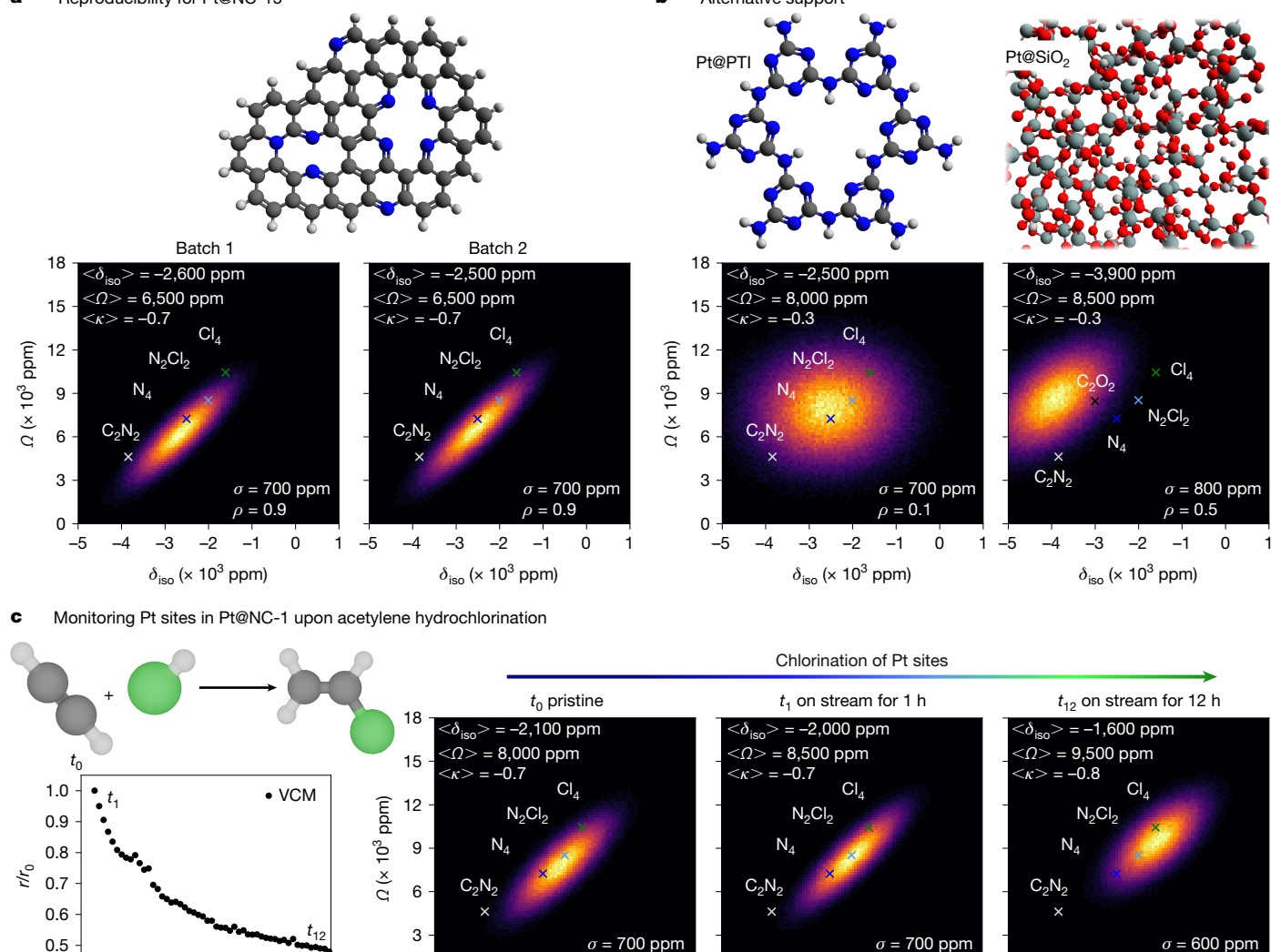

**a** Reproducibility for Pt@NC-15

**b** Alternative support

Pt@PTI · Pt@SiO$_2$

**c** Monitoring Pt sites in Pt@NC-1 upon acetylene hydrochlorination

Chlorination of Pt sites

$t_0$ pristine · $t_1$ on stream for 1 h · $t_{12}$ on stream for 12 h

**Fig. 4 | Reproducibility, alternative supports and evolution upon catalysis by NMR signatures. a,b,** $\delta_{iso}$–$\Omega$-sampling maps with NMR signatures for two different batches of Pt@NC-15 (**a**) and PTI and SiO$_2$ supports (**b**), with illustrations of the structural fragments (C in black, N in blue, Si in grey and O in red) in the respective top panels. In the $\delta_{iso}$–$\Omega$-sampling map for Pt@SiO$_2$, the experimental data point[12] for a C/O-mixed Pt environment is additionally given for reference. **c,** Monitoring the hydrochlorination of acetylene with Pt@NC-1 as a function of time on stream at $t_0$ (pristine), $t_1$ (after 1 h) and $t_{12}$ (after 12 h). Normalized vinyl chloride monomer formation rate versus time on stream are shown in the left panel. $\delta_{iso}$–$\Omega$-sampling maps with NMR signatures for $t_0$, $t_1$ and $t_{12}$ are shown in the right panels. All $^{195}$Pt NMR spectra are shown in Extended Data Figs. 8 and 9.

are driven by the nature of atoms/ligands bound to Pt, clearly show that the immediate Pt(II) chemical environment in Pt@NC-5 after the first annealing step (Fig. 2e, −1,800 ppm/9,500 ppm) remains dominated by Cl (compare with K$_2$PtCl$_4$, Fig. 2a). After the second annealing, $<\delta_{iso}>$/$<\Omega>$ decrease notably (Fig. 2f, −2,300 ppm/7,500 ppm), documenting the change of the average coordination environment towards a Cl/N-mixed ligand set (compare with *cis*-Ptpy$_2$Cl$_2$, Fig. 2b), consistent with the removal of Cl during this treatment. For Pt@NC-15, we observed the same effect: although here the analysis shows that the average square-planar Pt(II) is already present in a mixed Cl/N environment after the first annealing step (Fig. 2g, −2,300 ppm/7,500 ppm) and Cl is further removed by the second temperature treatment, with Pt(II) in a N-dominated ligand set, moving even to Pt(II) directly bound to carbon (Fig. 2h, −2,600 ppm/6,500 ppm), paralleling the differences observed for *cis*-Ptpy$_2$Cl$_2$, Pt(NH$_3$)$_4$Cl$_2$ and even *cis*-Pt(Me)$_2$tmeda (Fig. 2b–d). These observations, in excellent agreement with Pt/Cl ratios

suggested by elemental analysis (Extended Data Table 1), provide more detailed information about the types and the distribution of elements/ligands bound to Pt.

Notably, the $\delta_{iso}$–$\Omega$-distribution maps (and $Q_{ICE}$ parameters) also effectively document the changes during annealing (Fig. 3c–f), with respective peak centres moving along the aforementioned line with slope $\Delta\Omega/\Delta\delta_{iso} = 3$. This shift goes from Cl-rich to Cl/N-mixed chemical environments for Pt@NC-5, and from Cl/N-mixed to N/C-mixed chemical environments for Pt@NC-15. The positive correlation coefficients ($\rho > 0$) found for all Pt@NC samples confirm a coordination-site continuum producing high-symmetry Pt(II) geometries. For Pt@NC-5, $\sigma$ and $\rho$ decrease upon further annealing from 600 to 400 ppm and 0.8 to 0.6, respectively. These changes indicate the beginning of the anchoring process of Pt onto the support, with Cl-removal (increase in chemical homogeneity, $\sigma$), and Pt binding onto N coordination sites (decrease of geometrical homogeneity, $\rho$). For Pt@NC-15, the increase of both $\sigma$

(from 400 to 700 ppm) and $\rho$ (from 0.7 to 0.9) confirms the almost full removal of Cl in the Pt(II) environment after the first annealing step and to the integration of Pt within the support upon the second annealing step. Anchoring is expected to yield a larger variety of coordination environments, induced by the intrinsic support inhomogeneity. In addition, the substantial decrease of $\delta_{iso}/\Omega$ indicates that Pt is also likely to be bound to carbon sites, as suggested by the DFT trends (Fig. 3b). The observed strong correlation between $\delta_{iso}$ and $\Omega$ testifies to a continuum of highly symmetric square-planar Pt(II) sites.

## Site quality across synthetic protocols

We next evaluated the reproducibility of Pt(II)-site structures in samples prepared following the same synthetic protocol. We prepared a second batch of a high Pt-content material (Pt@NC batch 2) containing 14 wt% and an identical Cl/Pt ratio as in Pt@NC-15 after the second annealing (Pt@NC batch 1). The comparison of NMR signatures for both samples (Fig. 4a and Extended Data Fig. 8a,b) clearly reflects close to identical Pt coordination environments. This is evidenced by the similar average CS-tensor and $Q_{ICE}$ parameters in both batches, demonstrating the reproducibility of this synthetic protocol, which is a typical challenge for catalyst development.

Another compelling application of our methodology is to compare Pt sites on an alternative polytriazine imide support (PTI, Fig. 4b, left panel and Extended Data Fig. 8c)[5]. The analysis of a Pt@PTI SAC with 2 wt% Pt revealed $<\delta_{iso}>/<\Omega> = -2,500/8,000$ ppm, suggesting Pt(II) coordination in an N-dominated to N/Cl-mixed chemical environment, which corroborates a previous conclusion[5]. However, unlike for NC supports, the $Q_{ICE}$ parameters for Pt@PTI reveal a more pronounced heterogeneity of the local geometry ($\rho = 0.1$), which is striking considering that PTI materials are semicrystalline and expected to present more uniform coordination sites. This increased heterogeneity could be related to several factors: (1) Pt coordination to both aminic and pyridinic N atoms, available with PTI support[5]; (2) presence of Li$^+$ ions known to be intercalated in PTI, (3) Pt integration within the layered structure of PTI; or (4) a partial Pt anchoring to PTI. Similarly, this methodology can be applied to oxide-supported materials; here illustrated with a single-site catalyst Pt@SiO$_2$, prepared by grafting (COD)Pt(OSi-(O$t$Bu)$_3$)$_2$ on silica. The average CS-tensor parameters ($<\delta_{iso}>/<\Omega> = -3,900/8,500$ ppm), that is, with the centre shifted from the $\Delta\Omega/\Delta\delta_{iso} = 3$ line, document a C/O-mixed chemical environment of the present Pt sites, in agreement with DFT computations (Fig. 3b and Extended Data Fig. 7) and previous reports[12,16], whereas $Q_{ICE}$ parameters indicate the presence of various local environments as expected for the amorphous silica support (Fig. 4b, right panel and Extended Data Fig. 8d).

## Catalytic reaction monitoring

Finally, we also demonstrated this approach to monitor the evolution of the local Pt-site structure in a Pt@NC SAC with 1 wt% Pt (Pt@NC-1) upon gas-phase acetylene hydrochlorination, which is a key reaction for the industrial production of vinyl chloride monomer[34]. We used Pt@NC SAC because the nitrogen-doped carbon carrier suffers from activity decay during time on stream so that our newly developed NMR methodology could obtain new insights about the deactivation process. Although Pt remains mostly dispersed as the catalyst deactivates according to HAADF-STEM imaging (Extended Data Fig. 3), we show that the NMR signatures evolve and that the chemical environment of Pt changes from N rich ($<\delta_{iso}>/<\Omega> = -2,100/8,000$ ppm) to Cl-rich ($<\delta_{iso}>/<\Omega> = -1,600/9,500$ ppm after 12 h on stream), which suggests that deactivation can be related to the change of coordination environment around Pt through excessive chlorination (Fig. 4c and Extended Data Fig. 9). This newly identified feature adds to the previously reported coking promoted by N sites[34], opening future

opportunities to uncover complex deactivation mechanisms in practically relevant processes.

Overall, readily accessible Pt NMR signatures provide unprecedented insights into Pt sites in SACs, elucidating their speciation and homogeneity. This quantitative analysis enables precise tracking of changes across synthetic steps, evaluating the reproducibility of a synthetic protocol and assessing Pt-site characteristics as a function of support material. Notably, we show that SACs, although having uniform nuclearity, display a distribution of coordination environments. Such rigorous analysis, previously inaccessible in catalysis and materials research, opens new avenues towards rational design strategies across SACs, as well as more complex atomically dispersed architectures, such as dual-atom and single-cluster catalysts. By advancing our ability to resolve coordination environments, this approach will facilitate the development of supports to anchor metal sites with well-defined local structures[9], advancing the field towards atomistic precision and bridging the gap with their homogenous analogues. Lastly, considering that most elements contain NMR-active isotopes, this approach can be extended across diverse catalytic materials and we are actively pursuing this direction.

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

## Methods

### Material synthesis

The nitrogen-doped carbon support was synthesized from a two-dimensional zeolitic imidazolate framework (2D-ZIF-8). The metal–organic framework was typically synthesized by dissolving 12.75 g zinc nitrate hexahydrate and 29.15 g 2-methyl imidazole separately in deionized water (DIW) (250 ml each). After mixing the two solutions, the mixture was stirred for 2 h and left still for 16 h. The resulting precipitate was filtered off, washed with DIW (3 l) and ethanol (1 l), and dried at 80 °C. The obtained 2D-ZIF-8 was mixed with KCl (20 g per g metal–organic framework), which was previously dissolved in water (0.5 g ml$^{-1}$). The water was removed in a rotary evaporator and then the solid was dried in a vacuum oven set a 80 °C. CN was obtained after carbonization at 700 °C (5 h, $N_2$ flow, 2 °C min$^{-1}$ heating rate) and a washing treatment consisting of hydrochloric acid (2 M in water), deionized water and ethanol followed by drying at 80 °C. Pt introduction was achieved using a wet impregnation method[32]. A target amount of chloroplatinic acid hexahydrate ($H_2PtCl_6.6H_2O$) was dissolved in 20 ml DIW and added to 250 mg NC. After sonication for 15 min, the water was evaporated and the powder dried at 80 °C overnight. The first annealing step was carried out at 200 °C (5 h, $N_2$ flow, 5 °C min$^{-1}$ heating rate). The catalyst was then suspended in a 1 M $NH_4Cl$ solution, sonicated for 30 min and removed by centrifugation. This process was repeated twice more with the 1 M $NH_4Cl$ solution and three more times with DIW. After another drying step, the material was annealed a second time at 550 °C (5 h, $N_2$ flow, 2 °C min$^{-1}$ heating rate). The Pt@PTI sample was reported in a previous study[5], from which Pt-PTI/773 was selected for the current work. The Pt@SiO$_2$ sample was prepared according to the procedure reported in ref. 12. Molecular compounds were either purchased ($K_2PtCl_4$, fluorochem, 99 %; Pt(NH$_3$)$_4$Cl$_2$.$x$H$_2$O, Merck, 98 %) or synthetized according to reported procedures (*cis*-Ptpy$_2$Cl$_2$; (tmeda)PtMe$_2$) (refs. 36–38).

### Catalytic evaluation

The hydrochlorination of acetylene (PanGas, purity 2.6) with HCl gas (Air Liquide, purity 2.8, anhydrous) was studied at atmospheric pressure. The gas flow (15 cm$^3$ STP min$^{-1}$) and composition (20 vol.% $C_2H_2$, 22 vol.% HCl, 8 vol.% Ar as an internal standard and 50 vol.% He as carrier gas) were controlled using digital mass flow controllers (Bronkorst). A quartz micro-reactor (10 mm inner diameter) equipped with a quartz frit was used to hold the catalyst bed (80 mg, particle size 0.2–0.4 mm). The catalyst bed was immobilized between two layers of quartz wool. The loaded reactor was equipped with bola valves and placed in a homemade electrical oven. Subsequently, the catalyst was heated to 200 °C under a pure helium flow. The reaction mixture was flown for a specified amount of time (1 h or 12 h) while the products were quantified online by means of gas chromatography–mass spectrometry (GC-MS, Agilent, GC 7890B, Agilent MSD5977 A, GS-Carbon PLOT column). Under the applied conditions, the initial acetylene conversion and vinyl chloride yield were 10.6% and 7%, respectively. These values ensured that the catalyst deactivation was evaluated in the low conversion regime. After the desired reaction time, the flow was reverted to pure helium and the reactor was cooled down under a pure helium flow. Subsequently, the bola valves were closed to prevent contact with air and moisture and the reactor was transferred into a glovebox (M Braun). The NMR rotor was packed within the glovebox before it was transferred to the spectrometer for measurement.

### Nitrogen physisorption

$N_2$ isotherms were recorded at −196 °C on a Micromeritics TriStar II analyser. Samples were degassed for 12 h at 200 °C under vacuum before measurement.

### Microscopy

HAADF-STEM at high magnifications was performed on a Jeol GrandARM operated at 300 kV. A Talos F200X instrument operated at 200 kV and equipped with an FEI SuperX detector was used for lower magnification micrographs in combination with energy-dispersive X-ray spectroscopy (EDX) for elemental mapping. Both microscopes are located in the ScopeM facilities of ETH Zürich in Zürich, Switzerland.

### Atom detection and nearest-neighbour analysis

Coordinates of Pt atoms in the micrographs were identified using a previously developed data-driven workflow[39], combining a convolution neural network and Gaussian mixture model. The analysis was done using the open-access app developed in the same work. An 80% confidence threshold was applied to predict the Pt atom locations in distinct particles, from which the distances to the nearest Pt neighbours were calculated. The results stemming from the different particles were combined to generate the overall distribution containing 747 locations and distances. The cut-off distance for multimer formation (0.22 nm) was used without modification, as the same materials were used in both studies.

### XPS

A Physical Electronics Quantera SXM instrument using monochromatic Al–Kα radiation was employed for XPS spectra. The Al–Kα radiation was generated from an electron beam operated at 15 kV, and the instrument was equipped with a hemispherical capacitor electron-energy analyser. The samples were analysed at an electron take-off angle of 45° and a constant analyser pass energy of 55 eV. The Pt 4$f$ spectra were fitted by mixed Gaussian–Lorentzian component profiles after Shirley background subtraction, whereas a pure Gaussian model was used for the N 1$s$ spectra. The peak positions are based on literature data and the NIST XPS database. The peak position error was fixed at ±0.2 eV. The detailed fitting parameters are given in Supplementary Tables 1 and 2. CasaXPS (v.2.3.23) was used for the XPS analysis[40].

### XAS

XAS experiments were performed at BM31 of the Swiss–Norwegian Beamlines (SNBL) located at the European Synchrotron Radiation Facility (ESRF) in Grenoble, France. Samples of Pt L$_3$ edge were collected in transmission mode using a double crystal Si (111) monochromator, and a secondary reference (Pt foil) was used for energy calibration (11,564.0 eV). Typical beam dimensions used were 0.4 mm (H) × 4 mm (W). The sample was packed into quartz capillaries (width 1.5 m, wall thickness 0.02 mm). Spectra were collected at beam energies ranging from 11.46 to −12.36 keV. The scans (five for molecular samples, ten for materials) were averaged to obtain a sufficient quality for structural analysis. Demeter software (0.9.24) from the Ifeffit software package (v.1.2.11) was used for the XAS data analysis[41]. The XAS data is shown Supplementary Figs. 1 and 2.

### NMR

$^{195}$Pt NMR measurements (static and 20 kHz MAS) on the molecular references, $K_2PtCl_4$, *cis*-Ptpy$_2$Cl$_2$, Pt(NH$_3$)$_4$Cl$_2$ and *cis*-PtMe$_2$tmeda, were conducted on a Bruker AVANCE III HD 400 spectrometer (9.4 T) at room temperature, equipped with a 3.2-mm Bruker MAS NMR probe. All compounds were finely ground and packed in regular-wall 3.2-mm zirconia rotors. $^{195}$Pt NMR measurements (static and 10 kHz MAS) on the samples containing single Pt atoms were performed on a Bruker AVANCE NEO 400 spectrometer (9.4 T) at low temperature (100 K), equipped with a 3.2-mm Bruker DNP MAS NMR probe. Thin-wall 3.2-mm zirconia rotors were employed. All static and MAS $^{195}$Pt NMR spectra were recorded using ultra-wideline NMR methodology[22], which employs advanced, frequency-swept pulse schemes

(wideband, uniform rate, smooth truncation[42]) for increased excitation bandwidths[23], and multiple-echo acquisition (Carr–Purcell–Meiboom–Gill[43,44]) for sensitivity enhancement. Recycle delays of 0.2 s were used throughout (Extended Data Fig. 1). For the molecular compounds cis-Ptpy$_2$Cl$_2$, Pt(NH$_3$)$_4$Cl$_2$ and cis-PtMe$_2$tmeda, additional $^1$H decoupling was used. The final $^{195}$Pt NMR spectra were obtained by coadding the individual echoes of the WCPMG-echo train in the time domain before Fourier transformation followed by magnitude calculation. In cases where the overall NMR linewidth exceeded the wideband, uniform rate, smooth truncation pulse bandwidth, subspectra were recorded, and the overall powder pattern was reconstructed using skyline projection. $^{195}$Pt chemical shifts were reported relative to K$_2$PtCl$_6$ (aqueous solution) set to 0 ppm. All further experimental details on the pulse sequences, experimental parameters and data processing are summarized in Supplementary Information section 2.

## NMR lineshape models

The $^{195}$Pt static and MAS NMR lineshapes for the molecular compounds K$_2$PtCl$_4$, cis-Ptpy$_2$Cl$_2$, Pt(NH$_3$)$_4$Cl$_2$ and cis-PtMe$_2$tmeda were computed using ssNake[45] based on the three input parameters $\delta_{iso}$, $\Omega$ and $\kappa$. For the computation of the $^{195}$Pt static and MAS NMR lineshapes of the SAC samples, we used two different approaches: (1) a Python routine to calculate fully converged lineshapes; and (2) a fast JavaScript code for spectral optimizations. The fully converged lineshapes were obtained by performing a Monte Carlo simulation using 10$^7$ individual CS tensors to establish the distribution of the $^{195}$Pt CS tensors based on the five input parameters $<\delta_{11}>$, $<\delta_{22}>$, $<\delta_{33}>$, $\sigma_{long}$ and $\sigma_{eq}$. These were binned in about 80.000–100.000 voxels, which were then summed up. The final $^{195}$Pt static and MAS NMR lineshapes were obtained by Fourier transform. Time-domain $^{195}$Pt NMR signals were presimulated using the SIMPSON simulations package[35,46], and the time required to simulate a fully converged lineshape was 1–2 min on a regular computer. For the fast lineshapes, an optimized Javascript module was developed for EasyNMR[47], which relied on a carefully selected subset containing only 1,000 of the CS tensors, and which provided the $^{195}$Pt NMR lineshapes in less than a second of computation time, with no visible difference between the fully converged lineshapes. A detailed description of the Python and Javascript routines, as well as a comparison of the numerical model presented in the main text and of existing, well-established protocols for computing NMR powder patterns that consider a site distribution (Czjzek[48] and extended model[33,49]) are provided in Supplementary Information sections 5.3 and 5.4.

## DFT computations

Geometry optimization of a series of Pt(II) model complexes and calculations of their $^{195}$Pt NMR spectroscopic parameters were performed using ADF 2022 with the hybrid PBE0 functional and Slater-type basis sets of quadruple-ζ quality for Pt, triple-ζ quality for the Pt first coordination shell and double-ζ quality for other atoms[50]. Relativistic effects were treated by the two-component zeroth order regular approximation[51–53]. Calculated isotropic chemical shift values were derived from the computed isotropic chemical shielding values by means of linear regression of experimental and computed values for a library of reference compounds (Supplementary Information section 4).

## Data availability

Raw $^{195}$Pt NMR data of SAC samples are available at Zenodo (https://doi.org/10.5281/zenodo.13381419)[54].

## Code availability

The program to compute the NMR lineshapes based on the numerical model introduced here is publicly available on https://easy.csdm.dk.

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

**Acknowledgements** This publication was created as part of NCCR Catalysis (180544 and 225147), a National Centre of Competence in Research funded by the Swiss National Science Foundation. J.K. is funded by the European Union's Horizon Europe research and innovation programme under the Marie Skłodowska-Curie grant no. 101111472 'ParaMAS'. A.V.Y. and C.C. gratefully acknowledge the ETH+ Project SynMatLab for funding. A.V.Y. also thanks the ETH Career Seed Grant for financial support. A.J.P., G.P. and A.L. acknowledge funding from the Agence Nationale de la Recherche (ANR-21-CE29-0010-01 'MatPNMR' and ANR-22-CE93-0006-01 'Pt-NMR'). This work is part of a project that has received funding from the European Union's Horizon 2020 research and innovation programme under grant no. 101008500 ('PANACEA'). D.G. and C.C. gratefully acknowledge the Swiss National Foundation for financial support (SNF grant no. 200021L_213070). S. Büchele is acknowledged for providing the Pt-PTI sample. The authors also gratefully acknowledge X. Zhou (ETH Zürich), D. Stoian and W. van Beek (ESRF, SNBL, BM31) for performing the XAS measurements. D.G. acknowledges C. Hansen (ETH Zürich) for discussions on XAS data analysis. M.E.U. thanks D. Faust Akl and F. Krumeich (ETH Zürich) for valuable discussions on and help with material synthesis and characterization as well as microscopy, respectively. ScopeM and the high-performance computing group at ETH Zürich is acknowledged for access to facilities. We also thank SwissCat+ for the use of the low-temperature NMR cabinet connected to a 400-MHz NMR spectrometer.

**Author contributions** J.K. and A.V.Y. developed and optimized the experimental NMR protocols with the support of G.P., A.L. and A.J.P. J.K., A.V.Y. and D.G. carried out the NMR experiments. J.K. processed the experimental NMR data. M.E.U. synthesized the Pt SACs, performed the basic characterization and catalytic testing, processed the XPS experiments, including data analysis. S.M. and M.E.U. conducted the Pt-SAC nearest-neighbour analysis. J.K. developed the numerical model to compute the NMR lineshapes with the support of A.J.P. and T.V. J.K. and T.V. implemented the numerical model. T.V. implemented the numerical fitting routine of the NMR lineshapes and computed the confidence intervals. D.G. performed the DFT calculations, synthesized the non-commercial molecular compounds and Pt@SiO$_2$ and performed the analysis of the XAS spectra. A.V.Y., S.M., J.P.-R. and C.C. designed the project. C.C. and J.P.-R. supervised the project. The initial draft was written by J.K., A.V.Y. and C.C. All authors contributed to the development of the final manuscript.

**Funding** Open access funding provided by Swiss Federal Institute of Technology Zurich.

**Competing interests** The authors declare no competing interests.

**Additional information**
**Correspondence and requests for materials** should be addressed to Javier Pérez-Ramírez or Christophe Copéret.

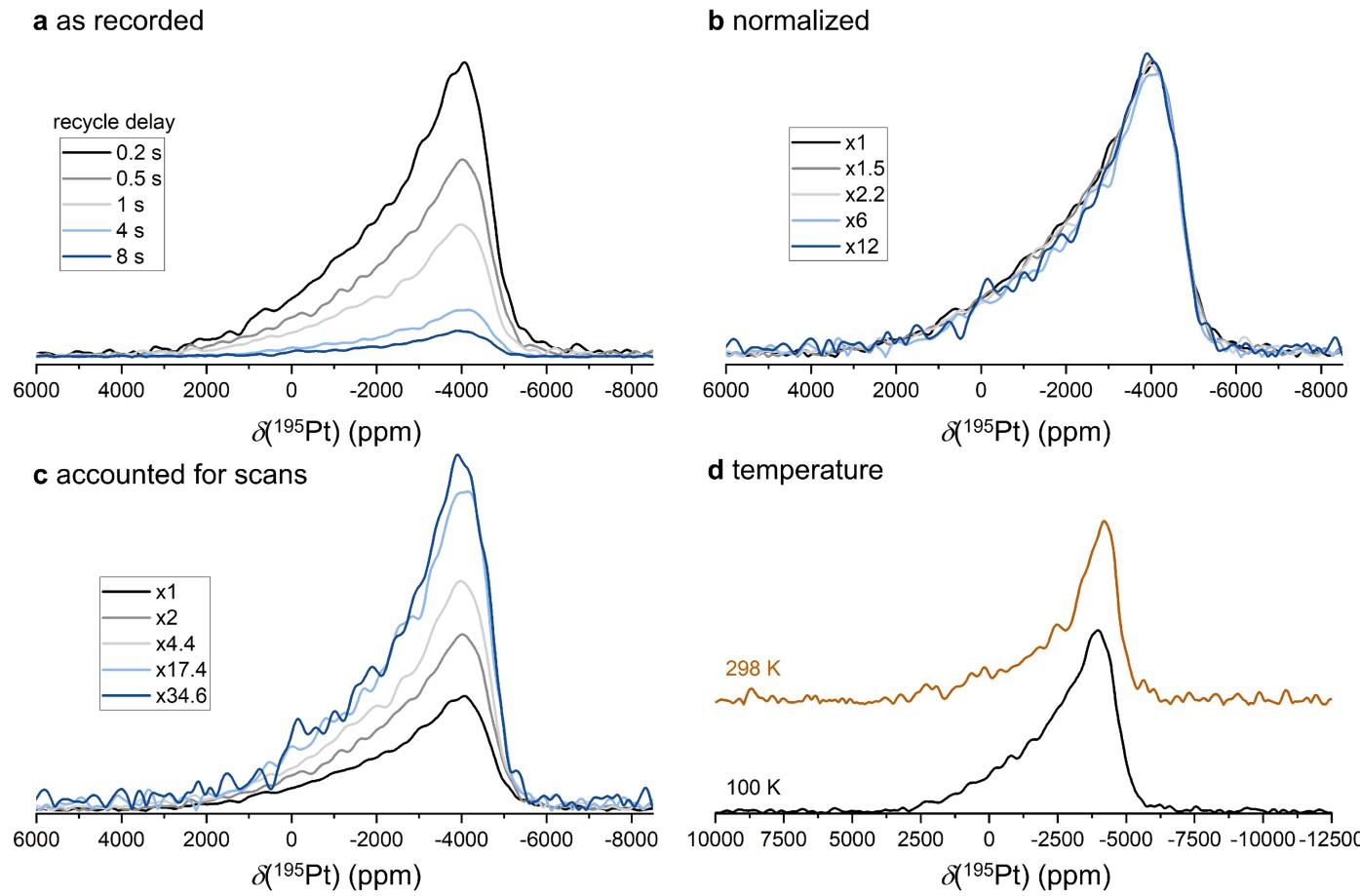

**a** as recorded

recycle delay
— 0.2 s
— 0.5 s
— 1 s
— 4 s
— 8 s

$\delta(^{195}Pt)$ (ppm)

**b** normalized

— x1
— x1.5
— x2.2
— x6
— x12

$\delta(^{195}Pt)$ (ppm)

**c** accounted for scans

— x1
— x2
— x4.4
— x17.4
— x34.6

$\delta(^{195}Pt)$ (ppm)

**d** temperature

298 K

100 K

$\delta(^{195}Pt)$ (ppm)

**Extended Data Fig. 1 | Recycle delay and temperature dependence of the $^{195}$Pt NMR signal.** Effect of the recycle delay at 100 K (a)-(c) and the temperature at a recycle delay of 0.2 s (d) on the static $^{195}$Pt NMR signature of Pt@NC-15 (batch 2). In (a), the experimental time for each spectrum has been kept approximately constant, amounting to 2.6 h (46560 scans for 0.2 s), 2.8 h (20300 scans for 0.5 s), 2.9 h (10470 scans for 1 s), 3 h (2680 scans for 4 s), and 3 h (1344 scans at 8 s), respectively. (b) Spectra shown in (a) normalized to unity. (c) Spectra shown in (a), rescaled to account for the different number of scans. In (d), the experimental time at RT has been increased by a factor of eight to ~ 20.5 h (368280 scans at 0.2 s recycle delay). For additional discussion, see Section S3 in the Supporting Information.

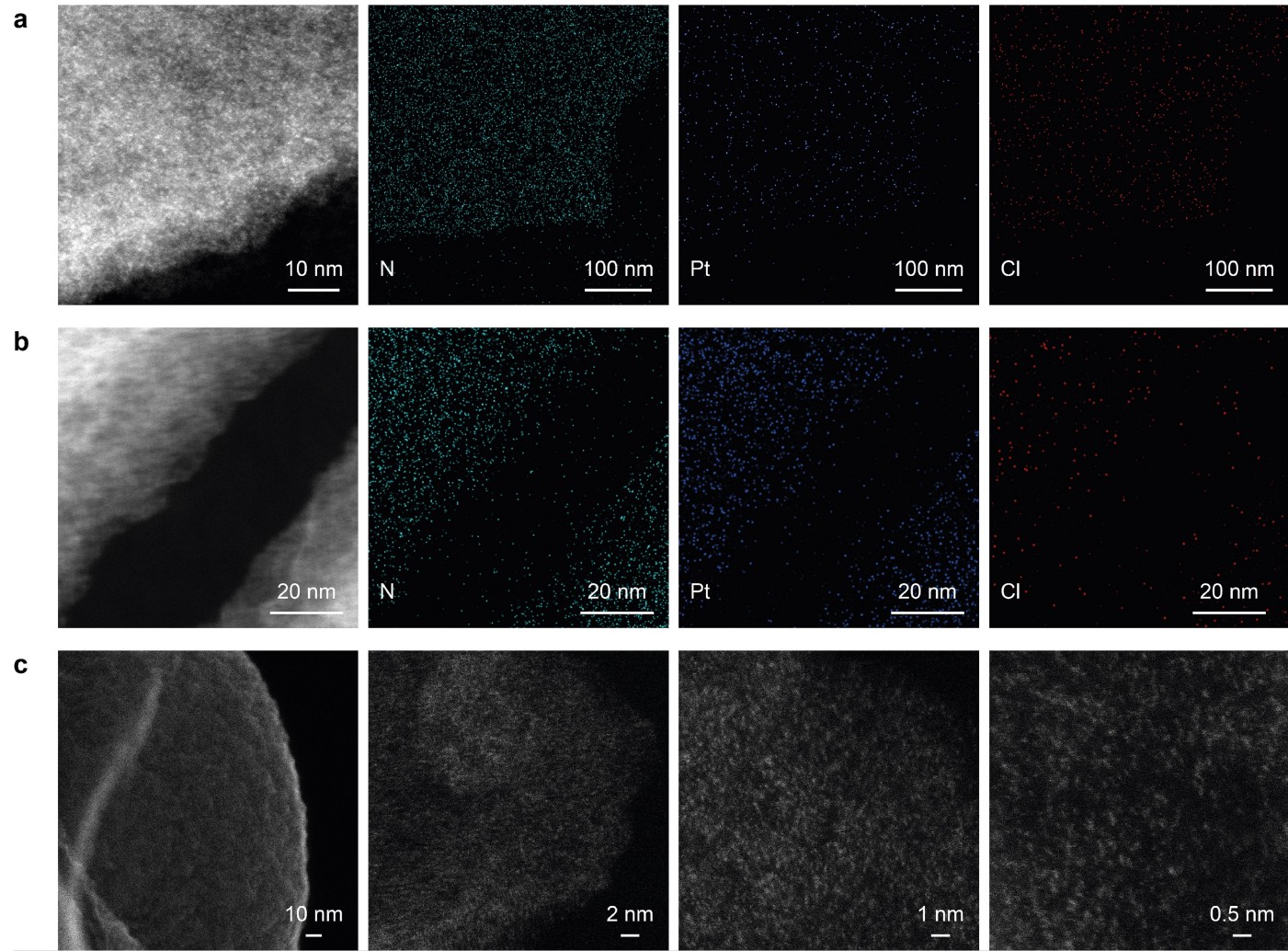

**Extended Data Fig. 2 | HAADF-STEM images with EDX maps.** (a) Pt@NC-5 and (b) Pt@NC-15 confirming the presence and dispersion of the elements of interest, (c) High-resolution micrographs of Pt@NC-15 after the second annealing step at different magnifications and sample regions, confirming atomic dispersion of Pt on the surface of NC.

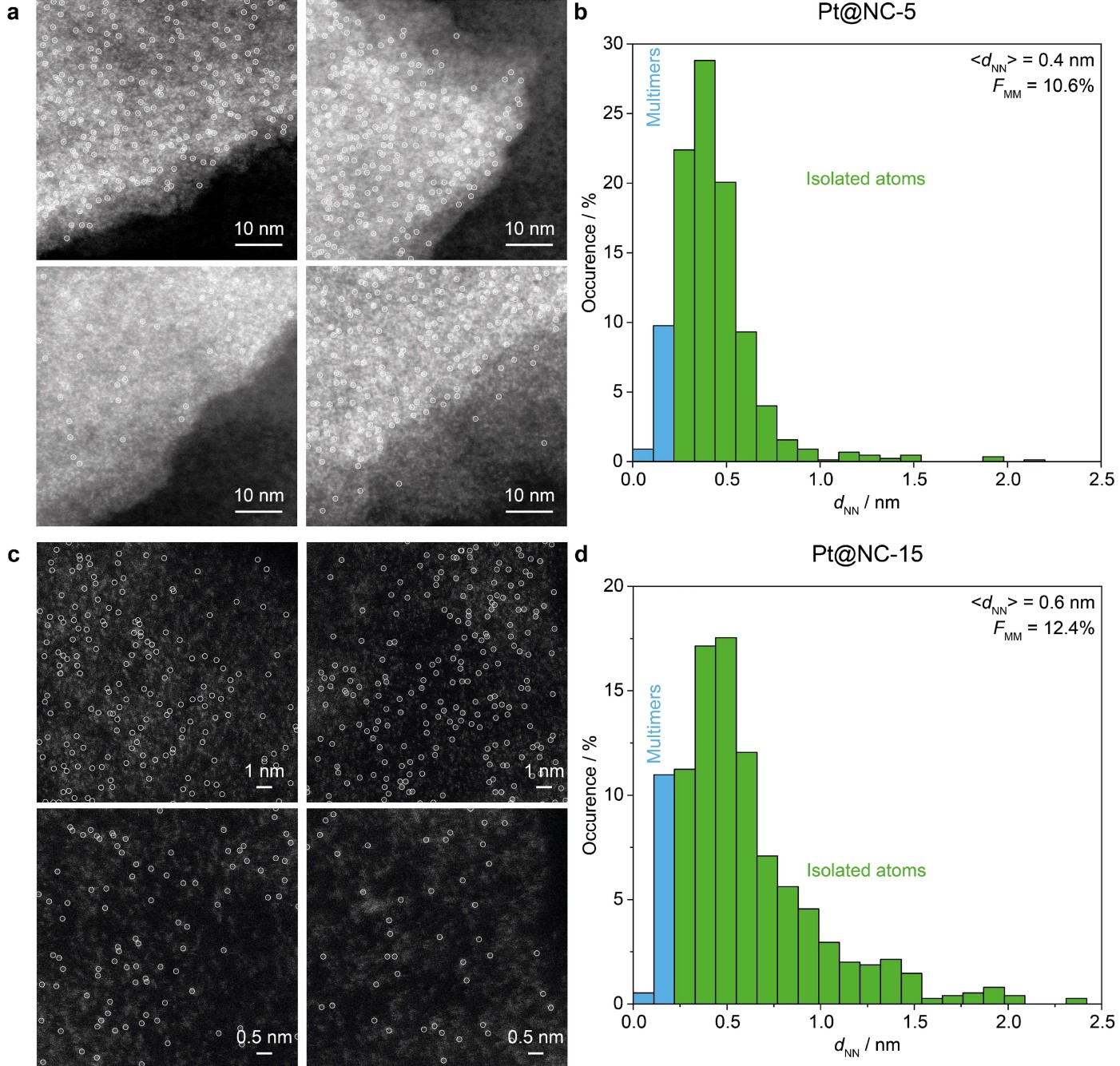

**Extended Data Fig. 3 | Automated atom detection analysis of HAADF-STEM images.** Pt@NC-5 (top) and Pt@NC-15 (bottom) after the second annealing step. (a) and (c) High-resolution STEM micrographs with machine learning detected atoms overlaid (white circles). (b) and (d) Distribution of the nearest-neighbor distance ($d_{NN}$) between detected metal sites. The estimated upper bounds for multimer formation are 10.6% and 12.4%, defined by atoms separated by <0.22 nm. As STEM provides 2D projections of 3D structures, this value is known to overestimate the actual degree of multimer formation. The analysis was performed using a previously developed open-access tool with a confidence threshold set to 80%.

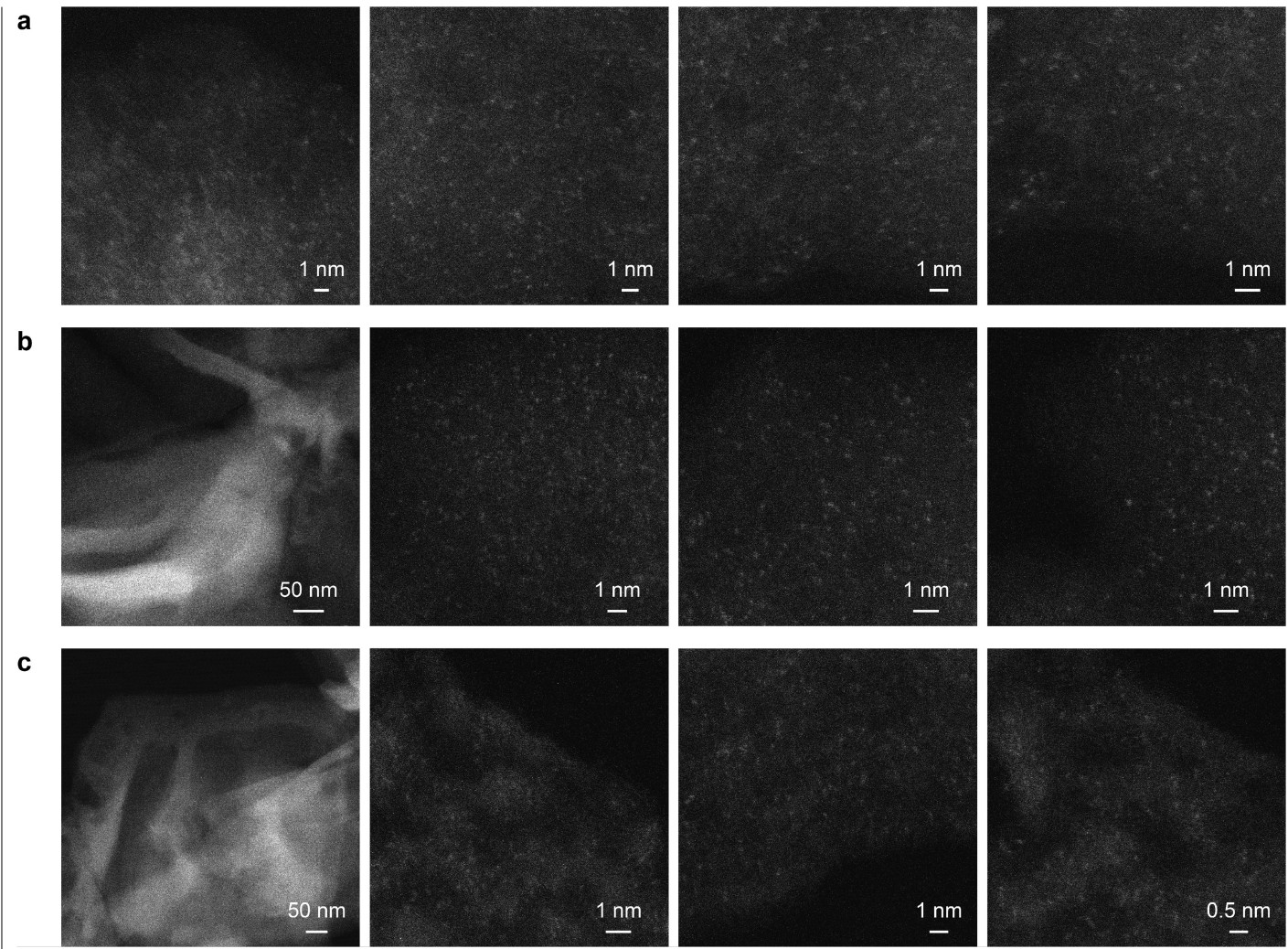

**Extended Data Fig. 4 | HAADF-STEM images of Pt@NC-1.** (a) after the second annealing step, (b) Pt@NC-1-1h and (c) Pt@NC-1-12h, which are the catalysts used in the hydrochlorination of acetylene after 1 h and 12 h on stream.

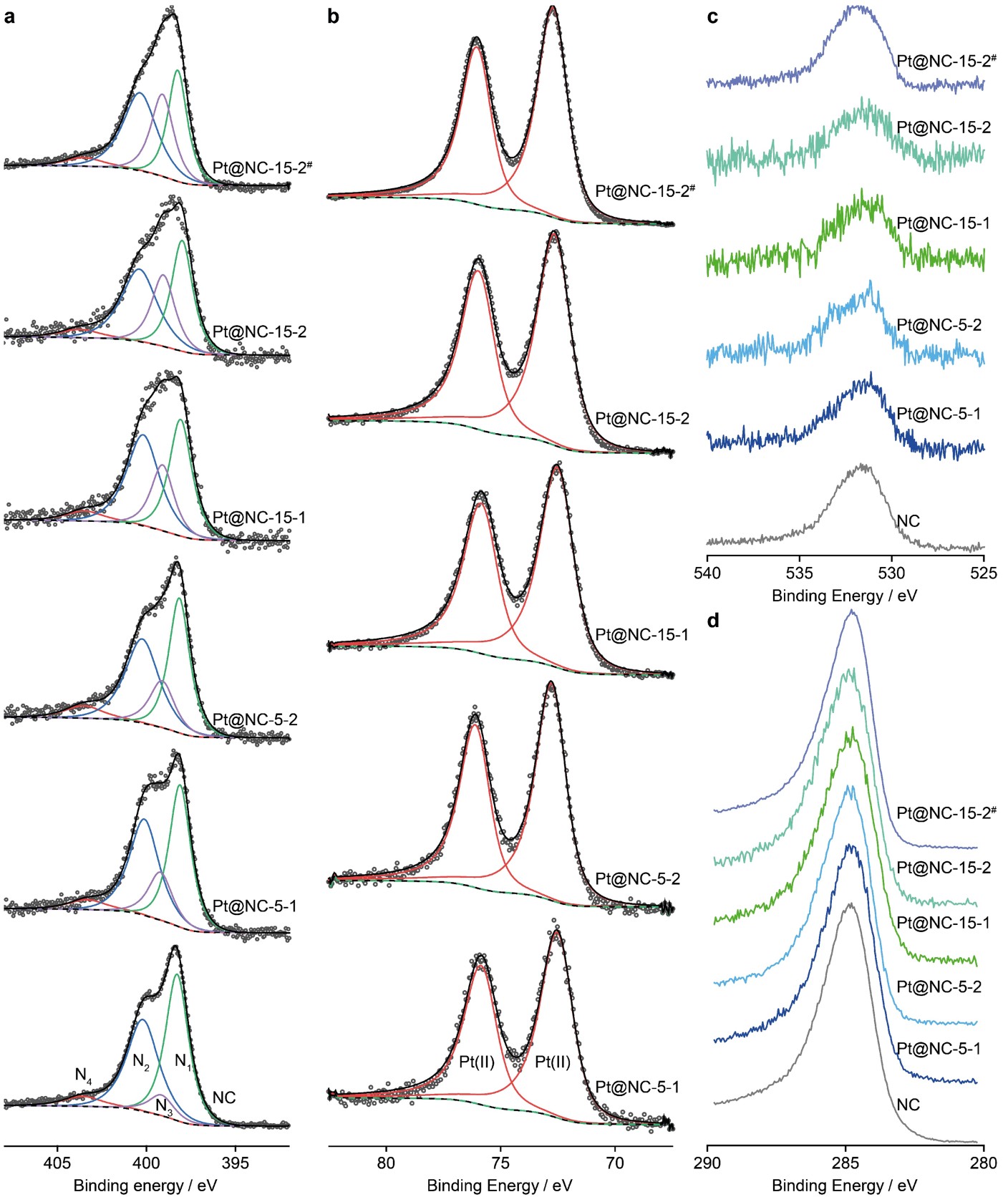

**Extended Data Fig. 5 | XPS data for the Pt@NC samples studied in this work.** N 1 s with fitting (a), Pt 4 f with fitting (b), O 1 s (c), C 1 s (d). In the fitted signal, the background is shown as a dashed line, the raw signals as circles, and the envelope as solid black lines. Code for (a): $N_1$: pyridinic N; $N_2$: pyrrolic N; $N_3$: metal-bound or amine N; $N_4$: oxidized N. In (b), Pt(IV) and Pt(0) contributions do not differ from the background. For additional discussion, see Section S1.3 in the Supporting Information.

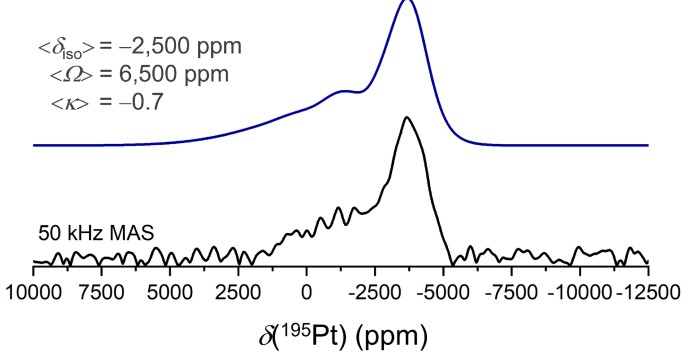

$\langle\delta_{iso}\rangle = -2,500$ ppm
$\langle\Omega\rangle = 6,500$ ppm
$\langle\kappa\rangle = -0.7$

50 kHz MAS

10000 7500 5000 2500 0 -2500 -5000 -7500 -10000 -12500

$\delta(^{195}Pt)$ (ppm)

**Extended Data Fig. 6 | $^{195}$Pt MAS NMR spectrum of Pt@NC-15 (batch 2) at 50 kHz MAS.** Experimental data include the first subspectrum. The model (blue line) was obtained based on the parameters used to compute the static lineshape shown in Fig. 4a in the main text. For additional discussion, see Section S3 in the Supporting Information.

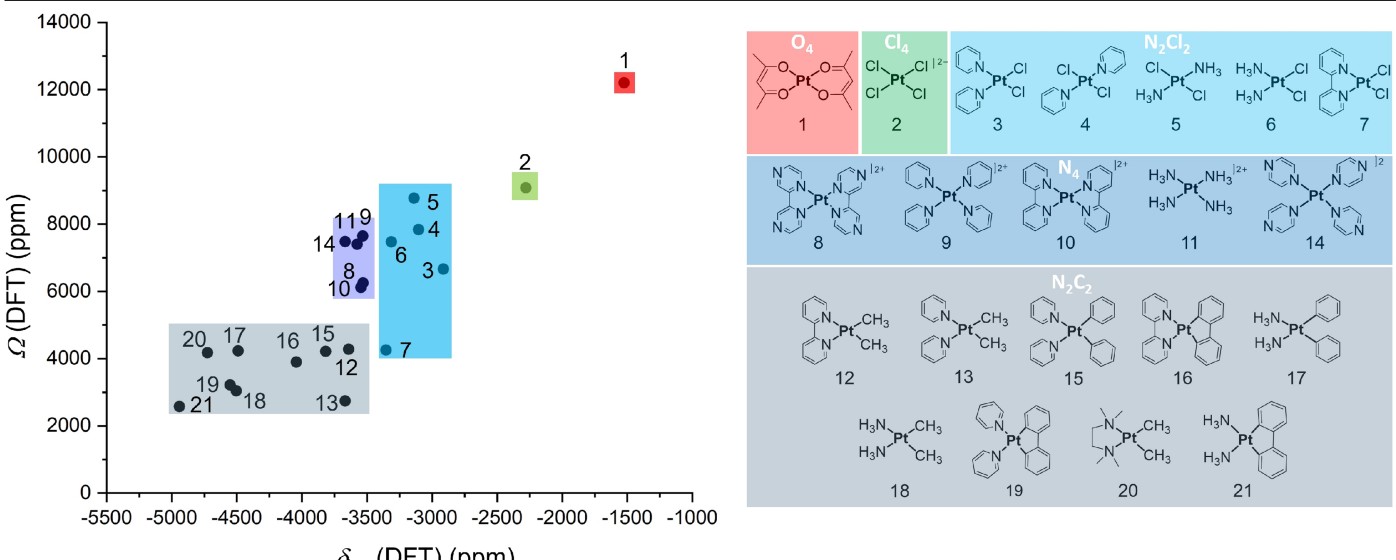

**Extended Data Fig. 7 | $\delta_{iso}$-$\Omega$ map for the calculated Pt(II) square-planar model complexes (1–21).** Compounds are grouped according to their coordination environment to the Pt atom (O$_4$, red; Cl$_4$, green; N$_2$Cl$_2$, light blue; N$_4$, blue; N$_2$C$_2$, grey). Structures are numbered with respect to (ascending) $\delta_{iso}$ values. For additional discussion, see Section S4 in the Supporting Information.

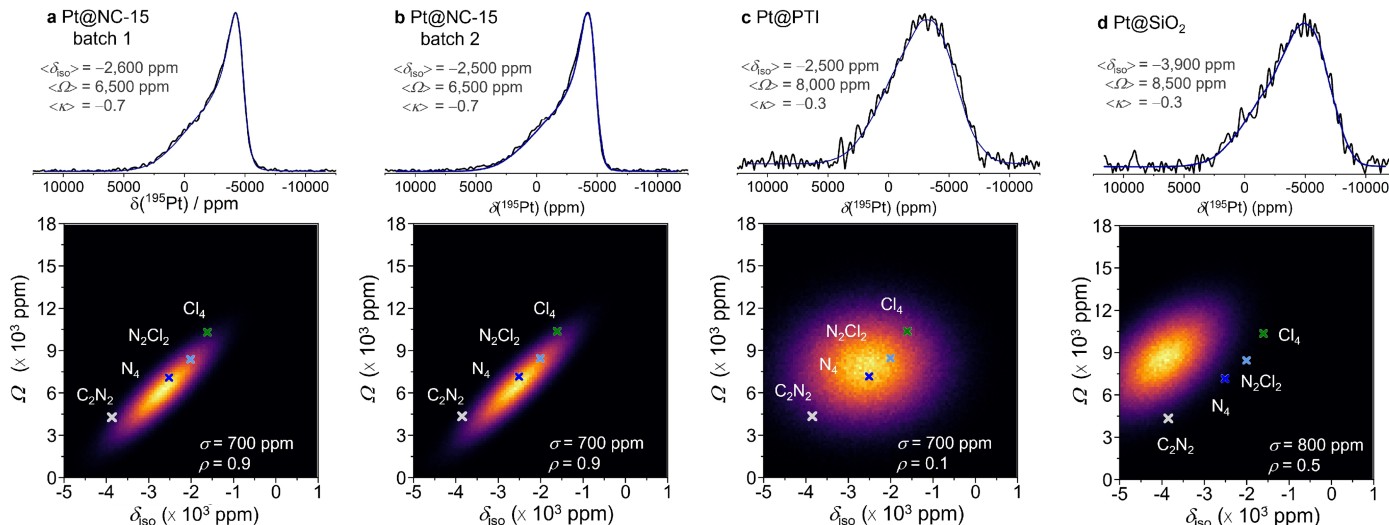

**Extended Data Fig. 8 | Static ¹⁹⁵Pt NMR signatures for reproducibility and alternative supports.** Pt@NC-15 after the second annealing step batch 1 (a) and 2 (b), Pt@PTI (c), and Pt@SiO₂ (d).

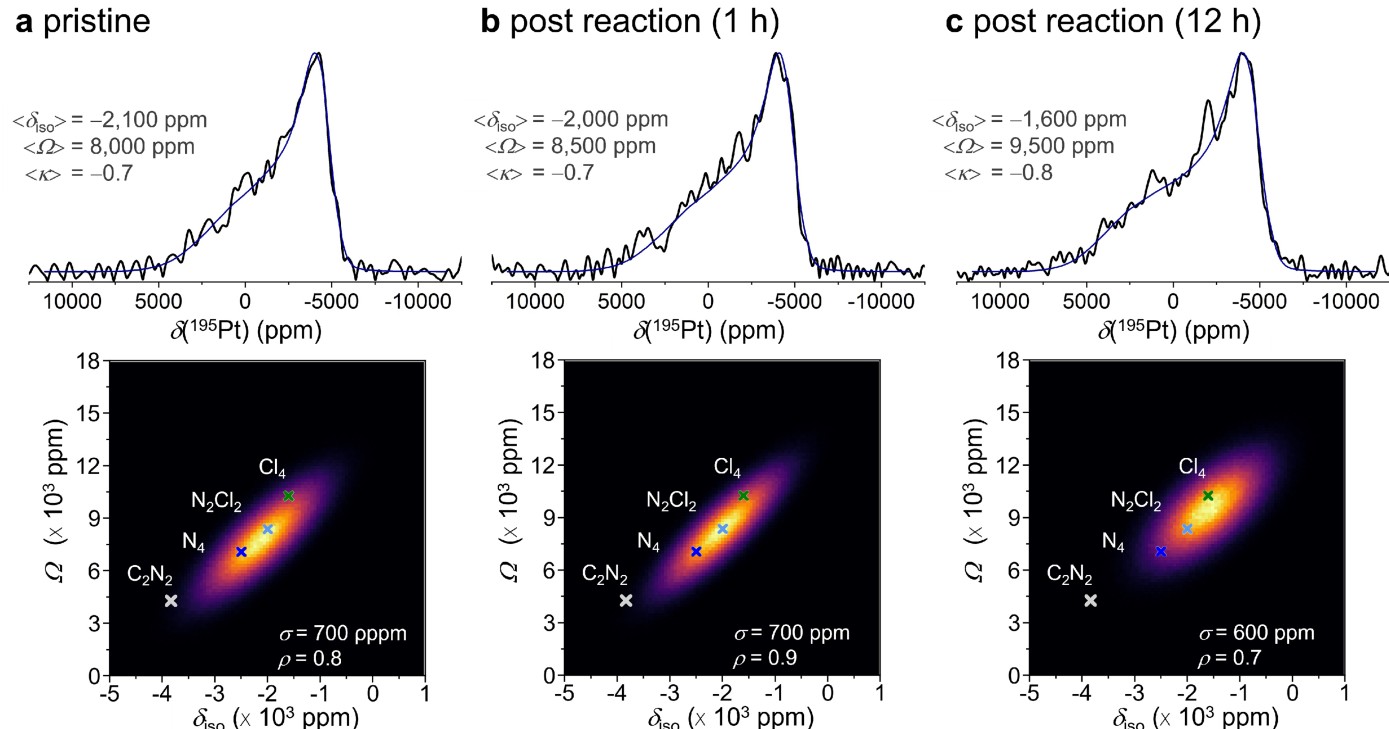

**a** pristine

$\langle\delta_{iso}\rangle$ = −2,100 ppm
$\langle\Omega\rangle$ = 8,000 ppm
$\langle\kappa\rangle$ = −0.7

**b** post reaction (1 h)

$\langle\delta_{iso}\rangle$ = −2,000 ppm
$\langle\Omega\rangle$ = 8,500 ppm
$\langle\kappa\rangle$ = −0.7

**c** post reaction (12 h)

$\langle\delta_{iso}\rangle$ = −1,600 ppm
$\langle\Omega\rangle$ = 9,500 ppm
$\langle\kappa\rangle$ = −0.8

**Extended Data Fig. 9 | Static $^{195}$Pt NMR signatures of Pt@NC-1 upon hydrochlorination of acetylene.** (a) the pristine catalyst, (b) after 1 h, and (c) after 12 h on stream.

**Extended Data Table 1 | EA composition of the Pt@NC materials and resulting Cl/Pt atomic ratios**

| Material | Ann. step | C (wt%) | H (wt%) | N (wt%) | Cl (wt%) | Pt (wt%) | Cl/Pt ratio |
|---|---|---|---|---|---|---|---|
| Pt@NC-5 | 1 | 53.24 | 1.92 | 21.55 | 2.83 | 4.33 | 3.60 |
| | 2 | 56.98 | 1.58 | 22.84 | 1.48 | 5.15 | 1.58 |
| Pt@NC-15 | 1 | 46.06 | 1.70 | 18.95 | 4.88 | 14.80 | 1.81 |
| | 2[#] | 50.23 | 1.27 | 20.29 | 2.37 | 14.60 | 0.89 |
| Pt@NC-15 (Pt@NC batch 2) | 2 | 52.16 | 1.38 | 18.02 | 2.28 | 13.90 | 0.90 |
| Pt@NC-1 | 2 | 58.17 | 1.68 | 20.59 | 2.33 | 1.2 | 10.69[$] |
| Pt@NC-1-1h* | 2 | 50.33 | 2.06 | 17.31 | 7.78 | 1.2 | 35.68[$] |
| Pt@NC-1-12h* | 2 | 51.3 | 1.87 | 17.25 | 7.94 | 1.1 | 39.79[$] |
| NC | - | 47.88 | 2.24 | 19.39 | 1.28 | - | - |

[#]In Fig. 4 in the main text also referred to as Pt@NC (batch 1) *Catalysts after being tested for 1 or 12h in the hydrochlorination of acetylene. [$]The Cl/Pt ratio is relatively high for the 1wt% Pt samples and exceeds the possible amount per Pt atoms, indicating chlorination of the support itself during synthesis and reaction (see Section S3 of the Supporting Information for further discussion).