## [Peer Review File · Nature]

Coordination environments of platinum single atom catalysts from NMR fingerprints

Corresponding Author: Professor Christophe Coperet

Version 0:

Reviewer comments:

Referee #1

(Remarks to the Author)

A long standing scientific challenge has been to accurately characterize the local chemical environment of single atom catalysts (SACs), such as those based on low amounts of Pt incorporated into a nitrogen-doped carbon support, and the manuscript by Koppe et al. provides a beautiful and direct approach to do so using ^{195}Pt NMR. Figure 1 documents state-of-the-art and the new strategy developed by Koppe et al. The reported solid-state NMR strategy takes advantage of recently developed ultra-wideline NMR techniques, WCPMG performed under static and MAS conditions, combined with low temperature (100 K) to obtain a ^{195}Pt NMR signal for the studied SACs (5-15 Pt wt%) within a reasonable time frame. The fact that ^{195}Pt in the studied SAC samples have very low T1 relaxation times (less than 0.2 s) also helps as explained in the SI. Figure 2 summarizes the obtained ^{195}Pt NMR spectra for a number of crystalline model compounds with different Pt(II) square planar coordination environments and the SACs with the compositions and annealing steps given in Table 1. Clearly, the ^{195}Pt NMR spectra performed under static and MAS for the models systems and studied SAC differ significantly and the authors make clear that the underlying reason for this is a δ_{iso} -distribution, exceeding the also applied MAS frequency of 50 kHz. Because of this, the challenge and novelty of the manuscript is how to setup a numerical model that (i) can describe these effects observed for the Pt-based SACs and (ii) relate the obtained parameters to structural features of SACs. To do so a numerical model is developed that accounts for the inhomogeneously broadened ^{195}Pt NMR signals due to the different Pt coordination environments present in SACs as described via distributions of the chemical shift (CS) tensor elements. The developed numerical model is explained and explored in detail in the SI and it enables a precise description of the dominant local Pt structure and site distribution based on Monte-Carlo simulations of five input parameters (d_{11} , d_{22} , d_{33} , σ_{long} , and σ_{eq}). This yields calculated average CS-tensor parameters δ_{iso} , Ω , and κ in addition to the quality index Q_{ICE} parameters σ and ρ , which together are considered as SAC NMR fingerprints. From the SI it is clear that the best representation of the numerical model is given by 2D Ω vs δ_{iso} contour plots, where the characteristic values of ρ and σ can be understood. A final ingredient is the DFT calculations of ^{195}Pt CS tensors for a range of model compounds (21 in total all with Pt(II) square planar coordination and various coordinations ligands). This gives the correlation in Figure 3b between $\Omega(\text{DFT})$ and $\delta_{\text{iso}}(\text{DFT})$ for ^{195}Pt (based on Figure S14), which provides a complete picture of the SAC fingerprints from ^{195}Pt NMR and the relevant coordination environments as used in Figure 3 and 4. To summarize, I find the work and strategy developed by Koppe et al. of very high quality with significant insights that was not possible before for Pt-based SACs. The manuscript is clearly written and is easy to follow and all experimental details are well documented. The references given are relevant and the number of references is appropriate. The SI contains (a lot of) additional information that clearly supports the material characterization, but it also explains the developed model for making distributions of the chemical shift (CS) tensor elements in detail. Thus, the work by Koppe et al. is in my opinion outstanding and merits publication in Nature. I recommend minor revision taking into account the comments below.

1. What is the pure Boltzmann gain when performing the ^{195}Pt NMR experiments at 100 K compared to ambient temperature, 298 K? Figure S8b seems to indicate only a moderate gain?
2. To me it was not clear how many offset spectra (or sub-spectra) were used for obtaining the full ^{195}Pt NMR spectrum of a given sample. Figure S11 seems to indicate two offsets? I would suggest the authors to include the employed offsets (in kHz) in Tables S5-S8.
3. Do the authors see any signs of different ^{195}Pt sites in the CPMG-echo trains after FT? Figure S9 gives a comparison of the 1st and 30th CPMG-echo. How does the 1st compare to the first 10 CPMG-echoes?
4. Did the authors consider analyzing the ^{195}Pt T2/T2' decays using Tikhonov regularization? See DOI: 10.1038/s41467-

020-14838-4 and DOI: 10.1021/acs.jpca.6b10007 for recent examples hereof. This approach might be an alternative way to show that the ^{195}Pt components for the studied SAC samples show site heterogeneity observed as a (broad) Gaussian distribution of ^{195}Pt isotropic chemical shifts.

5. I would recommend the authors to define Q_{ICE} and the associated quantitative quality index parameters in the context of the developed model described in the SI. Q_{ICE} is first introduced on p. S23 when describing the numerical protocol. It would be better if Q_{ICE} was introduced before in section 5.2 in the SI.
6. I would also like to encourage the authors to introduce more subsections in Section 5.2 to make it clearer what is being evaluated where. I believe this would make the model considerations even more accessible to the interested reader.
7. The authors should give a reference to Figure S14 in the caption of Figure 3 to indicate that the results shown in Figure 3b is derived from a broad range of compounds with Pt(II) square planar coordination.
8. A comment concerning the application the developed strategy to other metals (not necessarily $I = 1/2$) should be included in the final statements of the manuscript.
9. Labels for parts (d) to (f) are missing in Figure S7.

Referee #2

(Remarks to the Author)

This manuscript reports results and analyses for the use of ^{195}Pt solid-state NMR to evaluate the local environments of single-atom Pt sites in N-doped carbon heterogeneous catalysts. The manuscript is well written, well organized, and the topic is timely and important. The results are clearly presented and analyses are sound. The quantitative assessments of subtle differences in Pt coordination environments enabled by ^{195}Pt ssNMR provides detailed insights on the effects of interactions with ligands and local surface species that occur during synthesis treatments. The NMR data are analyzed in combination with Monte Carlo simulations and DFT calculations, as well as TEM, XPS, XAS, and EXAFS, which together yield detailed quantitative insights on the local structures and bonding environments, which are correlated with the preparation chemistry of single-atom Pt atoms supported on nitrogen-doped carbons. Furthermore, the approach can be expected to be general for other NMR-active nuclei with ultra-wide spectra.

There are several aspects that should be considered and addressed to improve the manuscript. First of all, the title refers to "catalysts" with no chemical reaction data provided for the materials investigated and only terse references to the literature within the Introduction. Rather, the manuscript reports changes to Pt sites during various treatments, but without specific relevance to catalytic properties that motivate the study. Inclusion of catalytic reaction data would increase the manuscript's impact. Second, because the single atom Pt-NC materials yield broad and relatively featureless ^{195}Pt ssNMR spectra that are similar in appearance (Figure 2e-h and Figure 4a-c), more discussion is needed with respect to the non-uniqueness and uncertainties associated with the lineshape analyses. It is reasonable that there are a distribution of support environments that account for the "blurred" and inhomogeneously broadened signals, as the authors note. But they are fit with parameters for which uncertainties are not provided and that, though different, cannot be considered unique. The Monte Carlo simulations and DFT calculations add confidence to the values obtained, however without uncertainty values for the chemical shift parameters and their effects on the correlation analyses, it is difficult to gauge whether the differences noted in the patterns reported in Figures 3 and 4 are significant. At several places in the manuscript the authors suggest that the relatively subtle differences in lineshape or the ν_{iso} vs. $\langle \text{iso} \rangle$ plots of Pt-coordination represent "fingerprints" or "signatures" of local Pt environments, but these appear to overstate the resolution and precision of the analyses and are hard to justify.

Several specific comments follow:

1. page 3, line 6 from bottom: Reference 21 appears to be incorrect where it is currently placed, as it does not feature ^{195}Pt ssNMR spectra with >1 month measurement times. Presumably the authors intended to cite the early pioneering references 27–29 by Slichter and coworkers. Reference 21 is still pertinent to the current manuscript, as it also reports single-atom Pt heterogeneous catalysts, which the authors should cite in this context and differentiate from their study.
2. page 4, line 5: The nitrogen-doped carbon supports are often used in electrochemical applications, because they are electrically conductive. It would be helpful to report the electrical conductivity values of the nitrogen-doped carbon materials used by the authors. The electrical conductivity of the support could result in Knight shift contributions to the ^{195}Pt lineshapes that would add another degree of complexity to analyses of local platinum environment. The authors should comment on this and justify whether the effects of support conductivity can be neglected.
3. page 4, Table 1: The high Pt contents of 5 wt% and 15 wt% are understandable for sensitivity reasons, though generally beyond what would normally be considered industrially relevant. Nevertheless, the relative uniformity of the single-atom catalysts, even at these high Pt loadings, may yield insights that are not be as dependent on Pt loading as for conventional catalysts based on supported Pt clusters. The authors should address whether or not this is the case and what effects Pt loading have on the distribution of Pt coordination environments.
4. page 5, Figure 2a-d: The ^{195}Pt ssNMR spectra of the reference compounds (Figure 2a-d) are useful, though are tersely discussed and without direct comparisons of each panel to the ^{195}Pt ssNMR spectra of the single atom catalysts. In addition, spectra of the reference compounds are shown for both static and magic angle spinning conditions, which except for the MAS spinning sidebands, appear to provide redundant information on the chemical shift tensor parameters. Without more clear relevance to the SACs, these spectra could be more suitably placed in the Supporting Information.
5. Page 5, Figure 2e-h: This is also the case for the single atom Pt catalysts, for which the respective ^{195}Pt ssNMR spectra acquired under static and MAS conditions appear indistinguishable and provide redundant information on the distributions of

chemical shift tensor parameters. Thus it seems unnecessary to include both the static and MAS spectra, and one set would be more suitably moved to the Supporting Information. In addition, it is not clear how the assignments of the $\langle \delta - 1 \rangle$ singularities in each spectrum are made, as their positions do not coincide with the approach of the intensity to the baseline at high-frequency. The relevances of the spectra of individual reference compounds selected for inclusion at the bottom of each panel are also not clear.

6. Pages 7-8, Figure 3: The combined use of the Monte Carlo simulations and DFT calculations, in conjunction with the experimental NMR data, are novel and powerful. The insights obtained represent the most impactful aspects of the manuscript. However, how the various parameters κ , σ , ρ , and $\langle \delta - \text{iso} \rangle / \Omega$ are arrived at and how the authors justify their physical significances are not clear in the main manuscript, because the important explanations are relegated to the extensive discussion in the S.I. For example, the authors assert conclusions about κ and $\langle \delta - \text{iso} \rangle / \Omega$ without explanation, which need more explanation to justify for a broad Nature readership.

7. Supporting Information, page S4, Figure S2: A specific description of each sample is needed in the caption. The labels are otherwise jargonish and obscure and make it difficult to cross compare with other figures. The resolution of the XPS survey scans is poor and lineshape fits are non-unique.

Overall, the strengths of the manuscript are notable: its combined Monte Carlo, DFT and NMR analyses provide important quantitative insights on the local structures and bonding environments that are correlated with the preparation chemistry of single-atom Pt atoms supported on nitrogen-doped carbons. Absent catalytic reaction data, the focus is on the authors' novel analytical approach to extracting and correlating chemical shift parameters and using them to probe Pt-coordination environments. Currently more discussion is needed concerning the effects of the uncertainties on the chemical shift parameters obtained from the broad ^{195}Pt NMR to justify the conclusions. And more discussion of the physical significances of the parameter correlations is needed to support the conclusions and to make the manuscript accessible to the Nature readership. If the authors do so, the manuscript will be improved and more suitable for publication in Nature.

Referee #3

(Remarks to the Author)

The ms by Copéret et al. reports new findings on the detailed characterization (coordination sphere of the metal) by NMR of some specific SACs. As specified in the introduction, such a level of characterization is essential to establish structure/property relationships; and it constitutes a lock to be lifted for the detailed understanding of these catalytic systems. The proposed methodology, which combine solid-state nuclear magnetic resonance spectroscopy and Monte-Carlo simulations, provides new insights into speciation and homogeneity for a given system: Pt on nitrogen-doped carbons. The study is conducted in a logical and appropriate manner, the results obtained are convincing and presented clearly. That being said, a certain number of points make me doubt the relevance of publishing this work in a very generalist inter-science and interdisciplinary journal like Nature.

1) It is noted that the characterization of surface-grafted coordination complexes by advanced NMR techniques has been known for more than 20 years (J. Am. Chem. Soc. 2001, 123, 16, 3820–3821). It is not specifically explained in the ms how SACs differ from these species and what makes them unique in terms of characterization.

2) Although the technique used is limited to NMR-active isotopes, only one system is presented; for example, there is no demonstration on Pt catalysts supported on oxides, an important system for catalytic converters. The study therefore seems to me to be very specific for an interdisciplinary journal.

3) As well stated in the abstract, such an approach should allow to establish structure/catalytic property relationships. The reader is frustrated not to find any. Moreover, such studies in modern catalysis today require in-situ or operando approaches (due to the dynamics of these systems). I have not found any comment from the authors concerning the applicability of the technique to in-situ conditions (the relatively long times for the analysis constitute a real handicap).

These considerations make me believe that this work, although of quality, should be published in a more specialized journal.

Version 1:

Reviewer comments:

Referee #1

(Remarks to the Author)

The authors have provided comprehensive and detailed responses to all of my raised questions, remarks, and suggestions for improving their work. Moreover, as suggested by Reviewer 2, the now included data of NMR spectroscopic monitoring of the catalytic performance for Pt@NC (1 wt%) in gas-phase hydrochlorination of acetylene (Figure 4c), really highlights the power of the developed approach as now insights about the catalyst deactivation processes can be made! Also, I find that the authors have made their model approach much clearer as suggested by both Reviewer 2 and myself. I congratulate the authors on their excellent work and I strongly recommend that their work should be accepted for publication in Nature.

Referee #2

(Remarks to the Author)

The authors have provided thoughtful responses to several of the comments and criticisms in the revised manuscript.

1. The addition of the subsections is helpful

2. The authors have connected their analyses of ^{195}Pt NMR spectra of heterogeneous platinum catalysts with reaction data in Figure 4 and in Figure S11 for the hydrochlorination of acetylene. This is a beneficial improvement, which increases the impact of the manuscript.

3. For the high Pt loadings investigated, it seems questionable that Pt atoms would only be present as single atoms. The authors describe their use of wet-impregnation to introduce Pt into an indistinctly described nitrogen-carbon support; impregnation often leads to aggregation of metals during activation treatments, in this case the annealing procedures, and especially for the high metal loadings of 5 wt% and 15 wt% reported. Such high Pt loadings also tend to obscure features in STEM images, which represent tiny regions that are insufficient to assert that the entire sample contains only isolated Pt atoms. Additional chemisorption or spectroscopic evidence should be provided to establish the extent of Pt dispersion to support the claim of single atom catalysts. That said, however, the emphasis on atomically dispersed Pt seems unnecessary, as the authors' analyses appear to be generally useful for assessing complicated inhomogeneous distributions of environments that would also be present if Pt aggregates are present. The authors' methodology would therefore seem to be suitable generally for analyses of inhomogeneous distributions of Pt environments, including in dispersed nanoclusters.

4. The brief mention of Figure 1 on page 3 is inadequate. The extensive information content of Figure 1 needs to be discussed or the figure should be condensed or omitted.

Figure 1b: The caption is incomplete, with no mention or description of the N or Cl tensor schematics or explanation of what the associated colors correspond to.

Figure 1c: The caption is incomplete with no description of what material the spectrum (left) corresponds to, 2D axis labels are missing (middle), and insufficient description of the significance of the different schematic diagrams (right, presumably sq. planar complexes with different ligands, grey is unclear) or their connection to the middle black square, which seems out of place.

5. Figure 2: The authors' response to the repetitive figure panels in Figure 2, while true, does not overcome the fact that the multiple similar spectra are insufficiently discussed and tedious for the reader to have figure out on one's own. Given the terse discussion, little is gained by including so many of them. I recommend that representative examples be included and discussed, referring to the others placed in the Supporting Information.

The authors should clarify how the values for $\langle \delta^{-1} \rangle$ are determined: they are not placed at the foot of their respective "blurred" lineshape where it reaches the baseline, but rather appear to be placed relatively arbitrarily where intensity is non-zero.

Describe "annealing" conditions in the figure caption at least briefly: in N_2 at 200 oC and then 550 oC.

6. Page 6: More detailed description is needed of the NC support: e.g., N content (from XPS data in Table S2) and porosity.

7. Page 7, bottom: What is the nature of Pt bonding with the N atoms in the carbon support? Schematic diagrams of the local bonding structures that are indicated by the data and the changes that occur with annealing would be helpful to include.

8. Figure 3: Explain why the 2nd annealing treatment narrows the heterogeneous distribution(s) in Fig. 3d, whereas annealing leads to broader features in Fig. 3f. Could the latter be due to Pt aggregation into clusters? If not, how can that be ruled out?

9. Figure 4: Reproducibility is necessary and presumed. The new results don't belong in the main text; Fig. 4a and its short discussion are more appropriate for the Supporting Information.

It is not clear what atoms Pt, N, C, O, H, Cl, etc. are represented in the 3 structures; all types of atoms need to be explicitly defined, especially those associated with Pt and its ligands.

10. Page 10: The catalytic reaction results are an important addition to the manuscript. The connection to catalyst deactivation is interesting.

11. The claim that the distribution of parameters is a "fingerprint" of molecular composition is an overstatement. Whereas fingerprints are unique to a given individual, this cannot be said (or at least has not convincingly been shown) for the combination of broad features that the authors' analytical approach determines for different catalysts. Based on the discussion provided, the uniqueness of the values obtained from such overlapping spectral features seems doubtful. This does not diminish the importance of the quantitative insights obtained on these challenging platinum catalysts, but use of "fingerprint" seems exaggerated and does not seem justified.

Overall, the strengths of the manuscript remain notable: its combined Monte Carlo, DFT and NMR analyses provide important quantitative insights on the local structures and bonding environments that are correlated with the preparation chemistry and catalytic reaction properties of Pt supported on nitrogen-doped carbons. The manuscript is improved, but the authors should consider the above comments before publication in Nature.

(Remarks to the Author)

I appreciate the effort made by the authors to extend the scope of supports, at least to one oxide (SiO₂) and especially to introduce chemical reactivity results even if in ex situ conditions (hydrochlorination of acetylene). I think that these important additions can contribute to a better impact of their work.

Referees' comments:

Referee #1 (Remarks to the Author):

A longstanding scientific challenge has been to accurately characterize the local chemical environment of single atom catalysts (SACs), such as those based on low amounts of Pt incorporated into a nitrogen-doped carbon support, and the manuscript by Koppe et al. provides a beautiful and direct approach to do so using ^{195}Pt NMR. Figure 1 documents state-of-the-art and the new strategy developed by Koppe et al. The reported solid-state NMR strategy takes advantage of recently developed ultra-wideline NMR techniques, WCPMG performed under static and MAS conditions, combined with low temperature (100 K) to obtain a ^{195}Pt NMR signal for the studied SACs (5-15 Pt wt%) within a reasonable time frame. The fact that ^{195}Pt in the studied SAC samples have very low T_1 relaxation times (less than 0.2 s) also helps as explained in the SI. Figure 2 summarizes the obtained ^{195}Pt NMR spectra for a number of crystalline model compounds with different Pt(II) square planar coordination environments and the SACs with the compositions and annealing steps given in Table 1. Clearly, the ^{195}Pt NMR spectra performed under static and MAS for the models systems and studied SAC differ significantly and the authors make clear that the underlying reason for this is a Δ_{iso} -distribution, exceeding the also applied MAS frequency of 50 kHz. Because of this, the challenge and novelty of the manuscript is how to setup a numerical model that (i) can describe these effects observed for the Pt-based SACs and (ii) relate the obtained parameters to structural features of SACs. To do so a numerical model is developed that accounts for the inhomogeneously broadened ^{195}Pt NMR signals due to the different Pt coordination environments present in SACs as described via distributions of the chemical shift (CS) tensor elements. The developed numerical model is explained and explored in detail in the SI and it enables a precise description of the dominant local Pt structure and site distribution based on Monte-Carlo simulations of five input parameters ($\langle d_{11} \rangle$, $\langle d_{22} \rangle$, $\langle d_{33} \rangle$, σ_{long} , and σ_{eq}). This yields calculated average CS-tensor parameters $\langle \Delta_{\text{iso}} \rangle$, $\langle \Omega \rangle$, and $\langle \kappa \rangle$ in addition to the quality index Q_{ICE} parameters σ and ρ , which together are considered as SAC NMR fingerprints. From the SI it is clear that the best representation of the numerical model is given by 2D Ω vs Δ_{iso} contour plots, where the characteristic values of ρ and σ can be understood. A final ingredient is the DFT calculations of ^{195}Pt CS tensors for a range of model compounds (21 in total all with Pt(II) square planar coordination and various coordination ligands). This gives the correlation in Figure 3b between $\Omega(\text{DFT})$ and $\Delta_{\text{iso}}(\text{DFT})$ for ^{195}Pt (based on Figure S14), which provides a complete picture of the SAC fingerprints from ^{195}Pt NMR and the relevant coordination environments as used in Figure 3 and 4. To summarize, I find the work and strategy developed by Koppe et al. of very high quality with significant insights that was not possible before for Pt-based SACs. The manuscript is clearly written and is easy to follow and all experimental details are well documented. The references given are relevant and the number of references is appropriate. The SI contains (a lot of) additional information that clearly supports the material characterization, but it also explains the developed model for making distributions of the chemical shift (CS) tensor elements in detail. Thus, the work by Koppe et al. is in my opinion outstanding and merits publication in Nature. I recommend minor revision taking into account the comments below.

We thank Reviewer #1 for their appreciation for our work and enthusiastically recommending the publication of our manuscript in Nature.

1. What is the pure Boltzmann gain when performing the ^{195}Pt NMR experiments at 100 K compared to ambient temperature, 298 K? Figure S8b seems to indicate only a moderate gain?

We thank Reviewer #1 for their comment. Figure S8b does not adequately reflect the Boltzmann gain, as we have extended the experimental time at RT by a factor of eight. At 100 K, we expect a S/N improvement by about a factor of three due to pure Boltzmann gain, corresponding to a reduction of the experimental time by a factor of nine. We have now clarified the experimental times for all spectra in the figure caption. We note that as we have further revised the figure and included new panels, the original Figure S8b is now called Figure S9d.

2. To me it was not clear how many offset spectra (or sub-spectra) were used for obtaining the full ^{195}Pt NMR spectrum of a given sample. Figure S11 seems to indicate two offsets? I would suggest the authors to include the employed offsets (in kHz) in Tables S5-S8.

Following the Reviewer's suggestion, we have revised the manuscript to include the number of offsets for all samples, see Tables S5-S9.

3. Do the authors see any signs of different ^{195}Pt sites in the CPMG-echo trains after FT? Figure S9 gives a comparison of the 1st and 30th CPMG-echo. How does the 1st compare to the first 10 CPMG-echoes?

Following the Reviewer's comment, we have now revised the SI and added in Figure S10 (former Figure S9) the lineshapes obtained from including the first 10 echoes, and the 10th echo only.

Figure S10. Effect of the CPMG acquisition on the static ^{195}Pt NMR signature of Pt@NC (batch 2). A comparison of the spectra resulting from FT and magnitude calculation of the echo obtained from adding up the first 100 WCPMG echoes (full echo train), the first 10 WCPMG echoes, as well as from the first, the 10th, and the 30th WCPMG echo only

Considering the low S/N in the spectra obtained from the individual echoes (in particular for the 10th and 30th), no significant changes in the ^{195}Pt NMR lineshapes are observed. The ^{195}Pt NMR lineshapes obtained from adding 10 and 100 WCPMG echoes can be considered identical, such that we do not observe clearly distinct ^{195}Pt sites.

4. Did the authors consider analyzing the ^{195}Pt T2/T2' decays using Tikhonov regularization? See DOI: 10.1038/s41467-020-14838-4 and DOI: 10.1021/acs.jpca.6b10007 for recent examples hereof. This approach might be an alternative way to show that the ^{195}Pt components for the

studied SAC samples show site heterogeneity observed as a (broad) Gaussian distribution of ¹⁹⁵Pt isotropic chemical shifts.

We thank Reviewer #1 for this comment. We indeed tried to analyze the transverse relaxation behavior using inverse Laplace transformation and Tikhonov regularization, even using a brand-new algorithm (manuscript under preparation) that includes the effect of the steady-state signal arising from the stimulated echoes in the present experimental setup. However, the relatively low S/N in each echo did not allow us to obtain high-quality inverse Laplace transformations for the studied SAC samples, and we could not detect differential T2 relaxation across the powder patterns (i.e., the presence of multiple sites).

5. I would recommend the authors to define $Q_{\{ICE\}}$ and the associated quantitative quality index parameters in the context of the developed model described in the SI. $Q_{\{ICE\}}$ is first introduced on p. S23 when describing the numerical protocol. It would be better if $Q_{\{ICE\}}$ was introduced before in section 5.2 in the SI.

6. I would also like to encourage the authors to introduce more subsections in Section 5.2 to make it clearer what is being evaluated where. I believe this would make the model considerations even more accessible to the interested reader.

We thank Reviewer #1 for these helpful comments. We followed their suggestion and have now revised the SI. We have introduced the following paragraphs in Section S5.2:

5.2.1 Representations of the distribution of CS tensors. (Page S21)

5.2.2 Correlation in the δ_{iso} - Ω -space. (Page S22)

5.3.3 The chemical-heterogeneity index σ . (Page S24)

5.3.4 The structural-heterogeneity index ρ . (Page S24)

We hope to have also addressed point 5, as the subheadings of the individual paragraphs help to clearly introduced the quality index in the context of section S5.2.

Based on the Reviewer's comment, we have made further minor modifications in Section S5 of the revised version of the SI, as marked in yellow. In particular, in subsection 5.2.2, we have added the exact expression for the correlation coefficient ρ , and an approximation which is valid for square-planar sites:

The correlation coefficient $\rho(\delta_{iso}, \Omega)$ can be calculated from the 10^7 individual CS tensors $\delta_{sq}^{(j)}$ according to

$$\rho(\delta_{iso}, \Omega) = \rho = \frac{\sum_j (\delta_{iso}^{(j)} - \langle \delta_{iso} \rangle) (\Omega^{(j)} - \langle \Omega \rangle)}{\sqrt{\sum_j (\delta_{iso}^{(j)} - \langle \delta_{iso} \rangle)^2 \sum_j (\Omega^{(j)} - \langle \Omega \rangle)^2}} \quad (1)$$

We note that if Eq. **Fehler! Verweisquelle konnte nicht gefunden werden.** is a valid approximation, we can likewise approximate the correlation coefficient, which is then given by

$$\rho \approx \sqrt{\frac{\sigma_{long}^2 - \sigma_{eq}^2}{\sigma_{long}^2 + 2\sigma_{eq}^2}} \quad (2)$$

where we have considered that $\Delta_{11}^{(j)}$, $\Delta_{22}^{(j)}$, and $\Delta_{33}^{(j)}$ are uncorrelated.

7. The authors should give a reference to Figure S14 in the caption of Figure 3 to indicate that the results shown in Figure 3b is derived from a broad range of compounds with Pt(II) square planar coordination.

Following the Reviewer's comment, we have now revised the manuscript to include a reference to Figure S17 (former Figure S14) in the caption of Figure 3.

8. A comment concerning the application the developed strategy to other metals (not necessarily $I = 1/2$) should be included in the final statements of the manuscript.

Following the Reviewer's comment, we have now included a statement at the end of the revised manuscript.

Lastly, considering that most elements contain NMR active isotopes, this approach can be extended to a broader family of SACs. We are actively working towards this direction.

9. Labels for parts (d) to (f) are missing in Figure S7.

We thank the Reviewer for pointing this out. We have included the missing labels in Figure S8 (former Figure S7) in the revised version of the SI.

Referee #2 (Remarks to the Author):

This manuscript reports results and analyses for the use of ^{195}Pt solid-state NMR to evaluate the local environments of single-atom Pt sites in N-doped carbon heterogeneous catalysts. The manuscript is well written, well organized, and the topic is timely and important. The results are clearly presented and analyses are sound. The quantitative assessments of subtle differences in Pt coordination environments enabled by ^{195}Pt ssNMR provides detailed insights on the effects of interactions with ligands and local surface species that occur during synthesis treatments. The NMR data are analyzed in combination with Monte Carlo simulations and DFT calculations, as well as TEM, XPS, XAS, and EXAFS, which together yield detailed quantitative insights on the local structures and bonding environments, which are correlated with the preparation chemistry of single-atom Pt atoms supported on nitrogen-doped carbons. Furthermore, the approach can be expected to be general for other NMR-active nuclei with ultra-wide spectra.

We thank Reviewer #2 for acknowledging the impact our work.

There are several aspects that should be considered and addressed to improve the manuscript. First of all, the title refers to “catalysts” with no chemical reaction data provided for the materials investigated and only terse references to the literature within the Introduction. Rather, the manuscript reports changes to Pt sites during various treatments, but without specific relevance to catalytic properties that motivate the study. Inclusion of catalytic reaction data would increase the manuscript’s impact.

We thank Reviewer #2 for this stimulating comment. In order to address it, we have decided to monitor the catalytic performance and the spectroscopic signature of Pt@NC (1 wt%) upon use in the gas-phase hydrochlorination of acetylene, a key reaction for the industrial production of vinyl chloride monomer. The full results are summarized in the new Figure 4c and S14 of the revised version of the manuscript and SI. We have now added a discussion regarding these results in the revised manuscript; it highlights the power of this methodology and shows that one can monitor deactivation processes and derive valuable structure-activity relationships.

Figure S11. Static ^{195}Pt NMR signatures, fitted lineshapes, and $\delta_{\text{iso}}-\Omega$ -distribution maps of Pt@NC-1 during hydrochlorination of acetylene for (a) the pristine catalyst, (b) after 1 h on stream, and (c) after 12 h on stream.

Figure 4. Reproducibility, alternative supports, and evolution during catalysis via NMR fingerprints. $\delta_{\text{iso}}\text{-}\Omega$ -sampling maps with NMR fingerprints for **a**, two different batches of Pt@NC-15, and **b**, PTI and SiO₂ supports, with illustrations of the structural fragments in the respective top panels. In the $\delta_{\text{iso}}\text{-}\Omega$ -sampling map for Pt@SiO₂, the experimental data point¹³ for a C/O-mixed Pt environment is additionally given for reference. **c**, Monitoring the hydrochlorination of acetylene with Pt@NC-1 as a function of time on stream at t_0 (pristine), t_1 (after 1 h), and t_{12} (after 12h). Normalized VCM formation rate vs time on stream in the left panel. $\delta_{\text{iso}}\text{-}\Omega$ -sampling maps with NMR fingerprints for t_0 , t_1 , and t_{12} in the right panels.

We have added the following discussion in the main manuscript:

Finally, we also demonstrated this approach to monitor the evolution of the local Pt-site structure in a Pt@NC SAC with 1 wt% Pt (Pt@NC-1) upon gas-phase acetylene hydrochlorination, a key reaction for the industrial production of vinyl chloride monomer (VCM)³⁶. We used Pt@NC SAC since the nitrogen-doped carbon carrier suffers from activity decay during time on stream and used our newly developed NMR methodology to obtain novel insights about the deactivation process. While Pt remains mostly dispersed as the catalyst deactivates according to microscopy (see **Figure S2**), we show that the NMR fingerprints evolve and that the chemical environment of Pt changes from N-rich ($\langle\delta_{\text{iso}}\rangle/\langle\Omega\rangle = -2100/8000$ ppm), to Cl-rich ($\langle\delta_{\text{iso}}\rangle/\langle\Omega\rangle = -1600/9500$ ppm after 12 h on stream), suggesting that deactivation can be related to the change of coordination environment around Pt through excessive chlorination (**Figure 4c**). This newly identified feature adds to the previously reported coking promoted by N sites³⁶, opening future opportunities to uncover complex deactivation mechanisms in practically relevant processes.

Second, because the single atom Pt-NC materials yield broad and relatively featureless ^{195}Pt ssNMR spectra that are similar in appearance (Figure 2e-h and Figure 4a-c), more discussion is needed with respect to the non-uniqueness and uncertainties associated with the lineshape analyses. It is reasonable that there are a distribution of support environments that account for the “blurred” and inhomogeneously broadened signals, as the authors note. But they are fit with parameters for which uncertainties are not provided and that, though different, cannot be considered unique. The Monte Carlo simulations and DFT calculations add confidence to the values obtained, however without uncertainty values for the chemical shift parameters and their effects on the correlation analyses, it is difficult to gauge whether the differences noted in the patterns reported in Figures 3 and 4 are significant. At several places in the manuscript the authors suggest that the relatively subtle differences in lineshape or the Ω vs. $\langle\delta_{\text{iso}}\rangle$ plots of Pt-coordination represent “fingerprints” or “signatures” of local Pt environments, but these appear to overstate the resolution and precision of the analyses and are hard to justify.

We thank Reviewer 2 for this very stimulating comment. We should note that while not discussed in the main text, the original submission contained an evaluation of the extracted parameters in the dedicated Section S5.5 *Robustness* in the SI, which was based on visual inspections of the calculated lineshapes, as we agree with the reviewer that the current model could be prone to local minima. Thus, it was necessary to be able to compute numerical lineshapes based on an optimization algorithm, rather than modeling the observed NMR lineshapes “by hand”. Such optimization would also allow us to compute confidence intervals for the input parameters, and with that demonstrate the existence of a global minimum in the five-dimensional input-parameter space to prove uniqueness of the reported parameters. With the previous way of computing lineshapes, in the present SI referred to as *fully converged lineshapes* (1-2 min per spectrum), we estimated that to require at least about one year of computational time. Therefore, we have in the last months put substantial efforts into speeding up the lineshape simulations, and we are now able to compute a full signature in about 900 ms. This involved using fast convergence of the numerical lineshapes and GPU-accelerated calculations. We have presented the details of the new procedure in the revised version of the SI (Section S5.4, Numerical protocol). We note that this routine is now publicly available on EasyNMR (<https://easy.csdm.dk>). With this acceleration, we were now able to compute numerically optimized lineshapes (real fits), involving the computation of up to 1000 lineshape per optimization. This has led to small changes in the reported output parameters; these changes have helped to refine the interpretation of the NMR signatures and the text has been edited accordingly.

Furthermore, we have chosen the sample Pt@NC-15 after the second annealing step to compute 95 % confidence intervals for the five input parameters. To this end, we have modified the original Section S5.5 *Robustness* to S5.5 *Uniqueness and stability*. Based on the 95 % confidence intervals, we have likewise calculated the 95 % confidence intervals for the output parameters. While we believe that the 95 % confidence intervals are very strict measure for the parameter uncertainties, we also acknowledge that the S/N varies across the samples, and we have decided to adjust the reported precision based on the confidence intervals for the output parameters. These values are all summarized in Table S12. We reprint this section here for the convenience of the reviewer.

5.5 Uniqueness and stability

With the availability of the very fast lineshape simulations described in the previous section, we are now able to further validate the numerical model proposed herein. To demonstrate the uniqueness of the reported sets of parameters, i.e., that they correspond to a global minimum in the five-dimensional parameter space, we have extended our numerical analysis to compute the 95 % confidence intervals for each of the input parameters $\langle\delta_{11}\rangle$, $\langle\delta_{22}\rangle$, $\langle\delta_{33}\rangle$, σ_{long} , and σ_{eq} . To do this, we first determined the optimum set of parameters $x_0 = (\langle\delta_{11}\rangle, \langle\delta_{22}\rangle, \langle\delta_{33}\rangle, \sigma_{\text{eq}}, \sigma_{\text{long}})_{\text{opt}}$ associated with the minimum value $\chi_0^2 = \chi^2(x_0)$. Then, one of the five input parameters was set to a fixed value, while the remaining four parameters were optimized to obtain the best fit (this parameter set is just labelled x), providing a minimum value of $\chi^2(x)$. The optimization is then repeated for various fixed values, covering a reasonable range of inputs. This is demonstrated in **Figure S23**, which shows the repeated optimization for different fixed values of (a) $\langle\delta_{11}\rangle$, (b) $\langle\delta_{22}\rangle$, (c) σ_{eq} , and (d) σ_{long} against the ^{195}Pt NMR spectrum of Pt@NC-15 after the second annealing step. Each of the black circles indicate the minimum value of χ^2 obtained after full optimization, where the global optimum parameter set x_0 is indicated in green. Clearly, in **Figure S23a, c, and d**, the circles describe a parabolic behaviour, that can be fitted using

$$\chi_{\text{est}}^2(x) = a(x - x_0)^2 + \chi_0^2, \quad (13)$$

shown as a blue line in **Figure S23**. In Eq. (13), a denotes the curvature, which can be used to calculate the 95 % confidence interval for the respective input parameter according to $C_{95\%} = 2/\sqrt{a}$.³⁰ All optimum parameters and 95 % confidence intervals are given in **Figure S23**. The confidence intervals are again summarized in **Table S12**.

Figure S23. 95 % confidence interval calculations. χ^2 values are reported for the fits of the ^{195}Pt NMR spectrum of Pt@NC-15 after the second annealing step. The spectrum was fitted based on the input parameters $\langle\delta_{11}\rangle$, $\langle\delta_{22}\rangle$, $\langle\delta_{33}\rangle$, σ_{long} , and σ_{eq} . Each data point (black circles) represents a fitting where the value indicated at the x axis is fixed at the specific value, but with the other parameters optimized. For all graphs, the black circles show the result of the fit for the given value of (a) $\langle\delta_{11}\rangle$, (b) $\langle\delta_{22}\rangle$, (c) σ_{long} , and (d) σ_{eq} . The solid blue lines represent the fit parabolic curve given in Eq. (13). The global optimum parameter set is labelled in green. In (b), the two parameters $\langle\delta_{22}\rangle$ and $\langle\delta_{33}\rangle$ are swapped at a value of around -4595 ppm as indicated by the arrow. For illustration, $\langle\delta_{33}\rangle$ resulting from the fitting is shown in red on the right y axis.

The analysis for the input parameter $\langle \delta_{22} \rangle$, demonstrated in **Figure S23b**, has a particular feature and requires special considerations. The optimum parameter set x_0 involves the first minimum observed at $\langle \delta_{22} \rangle = -4232$ ppm. Further increasing $\langle \delta_{22} \rangle$ from -4232 ppm towards 0 ppm shows a clear sharp edge as expected. Moreover, while a decrease of $\langle \delta_{22} \rangle$ generally results in higher values of χ^2 , a second minimum is found at $\langle \delta_{22} \rangle = -4946$ ppm. Upon forcing a decrease of $\langle \delta_{22} \rangle$, and optimizing $\langle \delta_{11} \rangle$, $\langle \delta_{33} \rangle$, σ_{long} , and σ_{eq} , the components $\langle \delta_{22} \rangle$ and $\langle \delta_{33} \rangle$ are swapped by the algorithm. To demonstrate this, the optimum value for the $\langle \delta_{33} \rangle$ component is shown in red in **Figure S23b**. This parameter varies linearly between the two minima. The local maximum at $\langle \delta_{22} \rangle = -4595$ ppm indicates the swap of the two components, and is indicated with the arrow. It is important to note that all parameter sets x enclosed by the two minima come in identical pairs, symmetrically flanking the local maximum at $\langle \delta_{22} \rangle = -4595$ ppm. These pairs describe the same average CS tensor, and of course result in identical NMR lineshapes. Therefore, the optimum parameter sets for $\langle \delta_{22} \rangle = -4232$ ppm and $\langle \delta_{22} \rangle = -4946$ ppm are likewise identical, and both corresponds to x_0 . We further point out, that **Figure S23b** generally reflects the high symmetry of the square-planar coordination for the local Pt(II) sites. While for the local maximum at $\langle \delta_{22} \rangle = -4595$ ppm, $\langle \delta_{22} \rangle = \langle \delta_{33} \rangle$, and thus $\langle \kappa \rangle = -1$, the optimum parameter set x_0 , where $\langle \kappa \rangle = -0.7$, corresponds to minor symmetry deviations on average. Any further deviations result in very high values of χ^2 , as demonstrated by the sharp parabolic edges above $\langle \delta_{22} \rangle = -4232$ ppm, and below $\langle \delta_{22} \rangle = -4946$ ppm. The best parabolic fit found according to Eq. (13) shows a compromise between these clear edges, and less sharp edge towards the local maximum at $\langle \delta_{22} \rangle = -4595$ ppm.

Overall, **Figure S23** clearly demonstrates the existence of a global minimum in the five-dimensional input-parameter space, that corresponds to a unique set of input parameters, and thus a unique numerical lineshape.

Based on the 95 % confidence intervals for the input parameters $\langle \delta_{11} \rangle$, $\langle \delta_{22} \rangle$, $\langle \delta_{33} \rangle$, σ_{long} , and σ_{eq} shown in **Figure S23**, we can now in principle likewise calculate the 95 % confidence intervals for the output parameters $\langle \delta_{\text{iso}} \rangle$, $\langle \Omega \rangle$, $\langle \kappa \rangle$, σ , and ρ . Assuming that there exists a function f_u , relating input parameters k and output parameters u , the 95 % confidence interval for an output parameter u can be calculated according to

$$C_{95\%}[f_u] = \sqrt{\sum_k \left(\frac{\partial f_u}{\partial x_k} \right)^2 C_{95\%}^2(x_k)}. \quad (14)$$

For the average CS-tensor parameters $\langle \delta_{\text{iso}} \rangle$, $\langle \Omega \rangle$, and $\langle \kappa \rangle$, the respective functions f_u are given in Eq. (5). We note that for $\langle \delta_{\text{iso}} \rangle$, the function $f_{\langle \delta_{\text{iso}} \rangle}$ according to Eq. (5) is exact, while Eq. (5) is a very close approximation for $\langle \Omega \rangle$ and $\langle \kappa \rangle$ due to potential re-ordering of the components $\Delta'_{11}(j)$, $\Delta'_{22}(j)$, and $\Delta'_{33}(j)$. Furthermore, there are no exact analytical functions f_u for σ and ρ . Yet, again, we can find very close approximations of f_σ by using the standard deviation of $\langle \delta_{\text{iso}} \rangle$ given in Eq. (6), and of f_ρ by using Eq. (9). The 95 % confidence intervals for all input and output parameters are summarized in **Table S12**.

Table S12. Summary of the 95 % confidence intervals and reported precision for input and output parameters.

	Parameter	95 % confidence interval	Reported precision
Input	$\langle \delta_{11} \rangle$	± 352 ppm	-
	$\langle \delta_{22} \rangle$	± 228 ppm	-
	$\langle \delta_{33} \rangle$	± 227 ppm	-
	σ_{long}	± 424 ppm	-
	σ_{eq}	± 167 ppm	-
Output	$\langle \delta_{\text{iso}} \rangle$	± 159 ppm	100 ppm
	$\langle \Omega \rangle$	± 419 ppm	500 ppm
	$\langle \kappa \rangle$	± 0.14	0.1
	σ	± 136 ppm	100 ppm
	ρ	± 0.07	0.1

While the 95 % confidence intervals do not directly represent the precision of the individual parameters, they merely represent statistical estimates of the likelihood of a given parameter. However, we find that the 95 % confidence intervals roughly match the precision of the parameters as based on the hundreds of optimizations and visual agreement between simulations and experiments. Hence, we use the 95 % confidence intervals to derive a reasonable precision for the parameters shown in the main text, as listed in **Table S12**. All exact and rounded parameters are summarized in **Table S13**.

We emphasize that the changes of parameters that we observe for different synthetic protocols, for different supports, and during the reaction remain notably larger than the reported precision, and thereby are clearly significant.

Several specific comments follow:

1. page 3, line 6 from bottom: Reference 21 appears to be incorrect where it is currently placed, as it does not feature 195Pt ssNMR spectra with >1 month measurement times. Presumably the authors intended to cite the early pioneering references 27–29 by Slichter and coworkers. Reference 21 is still pertinent to the current manuscript, as it also reports single-atom Pt heterogeneous catalysts, which the authors should cite in this context and differentiate from their study.

We thank Reviewer #2 for their comment. Reference 21 was included at this position, because this pioneering work had clearly estimated the challenge of carrying out these experiments (on page 2335, right column). Reference 21 discusses in particular a sample with 5 wt% Pt loading, and that a S/N comparable to that what they obtained for the 15 wt% sample would approximately require 36 days of experimental time, implicitly concluding that these experiments are therefore not obtainable. We did not include this reference when discussing single atom catalysts because the catalysts described in reference 21 are better described as Pt-sites bound to a coordination polymer, taking advantage of the well-defined coordination environment of these structures. In

contrast, SACs typically contain metal sites integrated in the material structure and involved annealing steps; while often expected as well-defined, our study highlight that they are indeed more heterogenous in nature.

2. page 4, line 5: The nitrogen-doped carbon supports are often used in electrochemical applications, because they are electrically conductive. It would be helpful to report the electrical conductivity values of the nitrogen-doped carbon materials used by the authors. The electrical conductivity of the support could result in Knight shift contributions to the ^{195}Pt lineshapes that would add another degree of complexity to analyses of local platinum environment. The authors should comment on this and justify whether the effects of support conductivity can be neglected.

We thank Reviewer #2 for the highly relevant comment. The main motivation to record the ^{195}Pt NMR signatures at different recycle delays (0.2s to 8s; see Figure S9a, former Figure S8a) was to clarify whether or not Korringa behavior is observed, as this would be a clear indicator for a Knight shift contribution. To make this clearer, we have revised Figure S9 and included a new representation of Figure S9a, where we have accounted for the different number of scans recorded for each recycle delay. This is now shown in Figure S9c. From this representation it is now immediately visible, that the ^{195}Pt NMR signal is saturated at recycle delays below ~ 4 s. Furthermore, we see that the saturation is uniform across the ^{195}Pt signature (see Figure S9b). Therefore, we have concluded that the Knight shift contribution is negligible, if present at all.

Figure S9. Effect of the recycle delay at 100 K (a)-(c) and the temperature at a recycle delay of 0.2 s (d) on the static ^{195}Pt NMR signature of Pt@NC (batch 2). In (a), the experimental time for each spectrum has been kept approximately constant, amounting to 2.6 h (46560 scans for 0.2 s), 2.8 h (20300 scans for 0.5 s), 2.9 h (10470 scans for 1 s), 3 h (2680 scans for 4 s), and 3 h (1344 scans at 8 s), respectively. (b) Spectra shown in (a) normalized to unity. (c) Spectra shown in (a), rescaled to account for the different number of scans. In (d), the experimental time at RT has been increased by a factor of eight to ~ 20.5 h (368280 scans at 0.2 s recycle delay).

We have included the following statement:

We note that since the nitrogen-doped carbon support is conductive, a hyperfine coupling between the conduction electrons of the support and the ^{195}Pt nuclei would potentially induce a ^{195}Pt Knight shift K .¹³ This Knight shift would sum to the chemical shift and therefore complicate the data analysis. This same hyperfine interaction would also give a contribution to the spin-lattice relaxation rate R_1 , which is related to the Knight shift via the Korringa relation,¹⁴ i.e., $K^2 \sim R_1$. Therefore, if the Knight shift does represent a significant contribution to the ^{195}Pt NMR lineshape, we would observe a correlated distribution of K and R_1 across the resonance. In particular, on reducing the recycle delay we would observe differential saturation of the signal intensity versus the shift. However, from **Figure S9b** and c, it is clear that the saturation we observe is uniform across the lineshape, and so there is no significant Korringa behavior, which further implies that the contribution from the Knight shift to the chemical shift is negligible, if present at all.

3. page 4, Table 1: The high Pt contents of 5 wt% and 15 wt% are understandable for sensitivity reasons, though generally beyond what would normally be considered industrially relevant. Nevertheless, the relative uniformity of the single-atom catalysts, even at these high Pt loadings, may yield insights that are not be as dependent on Pt loading as for conventional catalysts based on supported Pt clusters. The authors should address whether or not this is the case and what effects Pt loading have on the distribution of Pt coordination environments.

We thank Reviewer #2 for their comment. We note that in the initial submission included was a sample with 2 wt% Pt loading (Pt@PTI). Yet, we agree with this opinion, and hope to have addressed this concern with the inclusion of a Pt@NC sample with 1 wt% Pt loading, as discussed above.

4. page 5, Figure 2a-d: The ^{195}Pt ssNMR spectra of the reference compounds (Figure 2a-d) are useful, though are tersely discussed and without direct comparisons of each panel to the ^{195}Pt ssNMR spectra of the single atom catalysts. In addition, spectra of the reference compounds are shown for both static and magic angle spinning conditions, which except for the MAS spinning sidebands, appear to provide redundant information on the chemical shift tensor parameters. Without more clear relevance to the SACs, these spectra could be more suitably placed in the Supporting Information.

We appreciate the reviewer's perspective. The ^{195}Pt ssNMR spectra of the reference compounds are included to establish the link between the ^{195}Pt CS-tensor parameters and the local Pt environments, and we believe they are important in the context of the present study. Furthermore, while the static and MAS NMR spectra do of course provide identical information on the CS tensor, they in principle provide a different level of insight into the homogeneity of the Pt coordination environment. In case of molecular compounds, the narrow spinning sidebands are in particular important to emphasize the difference between well-defined and the more heterogeneous Pt sites in the SAC samples due to the different environments.

5. Page 5, Figure 2e-h: This is also the case for the single atom Pt catalysts, for which the respective ^{195}Pt ssNMR spectra acquired under static and MAS conditions appear indistinguishable and provide redundant information on the distributions of chemical shift tensor parameters. Thus it seems unnecessary to include both the static and MAS spectra, and one set would be more suitably moved to the Supporting Information. In addition, it is not clear how the assignments of the singularities in each spectrum are made, as their positions do not coincide with the approach of the intensity to the baseline at high-frequency. The relevances of the spectra

of individual reference compounds selected for inclusion at the bottom of each panel are also not clear.

Again, we appreciate the reviewer's comment. However, it is exactly the fact pointed out by the reviewer, i.e., that static and MAS spectra are identical, which provides convincing and undeniable experimental evidence for the presence of a (large) Pt site distribution in Pt SACs. In addition to that, the MAS spectra of the SAC samples allow us to obtain a lower limit of the extent of the Pt-site heterogeneity (comparison with MAS frequency), and are pertinent to legitimize the developed lineshape model. We also would like to emphasize that in the future, MAS spectra of SACs will be the methodology of choice to detect true single-site catalysts, i.e., SACs where all local Pt environments are (close to) identical.

Furthermore, based on the reviewer's comment, we have now in the revised version of Figure 2 removed the reference spectra in Figure 2e-h, as they indeed are not necessary for the discussion. Moreover, we added a sentence in the figure caption, stating that the indicated average tensor components are based on the input parameters used to obtain the best lineshape fit.

6. Pages 7-8, Figure 3: The combined use of the Monte Carlo simulations and DFT calculations, in conjunction with the experimental NMR data, are novel and powerful. The insights obtained represent the most impactful aspects of the manuscript. However, how the various parameters κ , σ , ρ , and Ω are arrived at and how the authors justify their physical significances are not clear in the main manuscript, because the important explanations are relegated to the extensive discussion in the S.I. For example, the authors assert conclusions about κ and Ω without explanation, which need more explanation to justify for a broad Nature readership.

We thank Reviewer #2 for their appreciation for the lineshape model in our work. The span (Ω) and skew (κ) are the well-known CS tensor parameters. We have in the original submission introduced the CS-tensor parameters including the mathematical expressions on page 3 of the main text:

“The NMR signature is associated with a characteristic lineshape, referred to as a powder pattern, which can be described by a chemical shift (CS) tensor, characterized by its three principal-axis components, $\delta_{11} > \delta_{22} > \delta_{33}$, whose average corresponds to $\delta_{iso} = 1/3(\delta_{11} + \delta_{22} + \delta_{33})$. The overall powder pattern linewidth is described by the span $\Omega = \delta_{11} - \delta_{33}$, which can vary from zero ppm for an isotropic Pt coordination, e.g., tetrahedral Pt(0)- or Pt(IV)-complexes, to more than 10'000 ppm for square-planar Pt(II), depending on specific ligand environments.^{14-17,13} Furthermore, the skew, $\kappa = 3(\delta_{22} - \delta_{iso})/\Omega$, describing the shape of the powder pattern¹⁹, contains information on the Pt-site symmetry with the minimum value $\kappa = -1$, reflecting the oblate square-planar Pt(II)-structural motif, and the maximum $\kappa = +1$ observed for prolate tensor shapes.”

Furthermore, all parameters are explained again in Figure 1 (with expression). Regarding the heterogeneity indexes σ and ρ , we have tried our best to find a suitable compromise: while we have indeed outsourced the derivation of these parameters to the SI (mostly due to the length restrictions for research articles in Nature), we have presented their interpretation (the physical significance, the index of heterogeneity in terms of chemistry and geometry) in an entire dedicated section “**Quantitative indexation of Pt coordination environments**” included in the main text of the original submission, where it is e.g., stated that (page 6):

“Both σ and ρ serve as a quantitative quality index for the Pt coordination environment (Q_{ICE}), describing heterogeneity in terms of chemical environment and local geometry, respectively.”

7. Supporting Information, page S4, Figure S2: A specific description of each sample is needed in the caption. The labels are otherwise jargonish and obscure and make it difficult to cross compare with other figures. The resolution of the XPS survey scans is poor and lineshape fits are non-unique.

We thank Reviewer #2 for their comment. We revised the labels of the samples in Figure S3 (former Figure S2) to make it clearer and more consistent with the labeling throughout the main text and the SI. We have also discussed the non-uniqueness of the lineshape fits explicitly.

Overall, the strengths of the manuscript are notable: its combined Monte Carlo, DFT and NMR analyses provide important quantitative insights on the local structures and bonding environments that are correlated with the preparation chemistry of single-atom Pt atoms supported on nitrogen-doped carbons. Absent catalytic reaction data, the focus is on the authors' novel analytical approach to extracting and correlating chemical shift parameters and using them to probe Pt-coordination environments. Currently more discussion is needed concerning the effects of the uncertainties on the chemical shift parameters obtained from the broad ^{195}Pt NMR to justify the conclusions. And more discussion of the physical significances of the parameter correlations is needed to support the conclusions and to make the manuscript accessible to the Nature readership. If the authors do so, the manuscript will be improved and more suitable for publication in Nature.

We thank Reviewer #2 for their appreciation for the strength of our work and generally suggesting its publication in Nature after addressing their comments. With the above-described revisions, we are positive to have addressed in sufficient depth the reviewer's main concerns.

Referee #3 (Remarks to the Author):

The ms by Copéret et al. reports new findings on the detailed characterization (coordination sphere of the metal) by NMR of some specific SACs. As specified in the introduction, such a level of characterization is essential to establish structure/property relationships; and it constitutes a lock to be lifted for the detailed understanding of these catalytic systems. The proposed methodology, which combine solid-state nuclear magnetic resonance spectroscopy and Monte-Carlo simulations, provides new insights into speciation and homogeneity for a given system: Pt on nitrogen-doped carbons. The study is conducted in a logical and appropriate manner, the results obtained are convincing and presented clearly.

We thank Reviewer #3 for acknowledging the importance of our work.

That being said, a certain number of points make me doubt the relevance of publishing this work in a very generalist inter-science and interdisciplinary journal like Nature.

1) It is noted that the characterization of surface-grafted coordination complexes by advanced NMR techniques has been known for more than 20 years (J. Am. Chem. Soc. 2001, 123, 16, 3820–3821). It is not specifically explained in the ms how SACs differ from these species and what makes them unique in terms of characterization.

We are not sure to understand this comment, which we think is somewhat misleading. We agree that earlier works have discussed the characterization of surface sites by solid-state NMR such as the reference pointed above by the reviewer and corresponding to one of the earlier work of Copéret *et al.*. However, the work described in this reference concerns the characterization by carbon-13 solid state NMR spectroscopy of a carbon-13 labeled organometallic compound grafted on silica, in other words the indirect characterization of well-defined and isotopically labeled surface sites by the NMR signature of its ligands. Such type of measurements has indeed become routine today in our group (and worldwide) and would certainly not qualify to be submitted to Nature today.

In great contrast, the present work detects the ^{195}Pt NMR signatures of highly dispersed Pt sites on N-doped carbon, a novel class of heterogenous catalysts, called single atom catalysts, and showing that the Pt signatures can directly probe the heterogenous environment of Pt due to the ill-defined nature of the support. We think it is indeed a major step that deserves attention beyond the realm of catalysis.

2) Although the technique used is limited to NMR-active isotopes, only one system is presented; for example, there is no demonstration on Pt catalysts supported on oxides, an important system for catalytic converters. The study therefore seems to me to be very specific for an interdisciplinary journal.

In the present manuscript, we described several materials and show that the preparation methodologies and the nature of the material (amorphous for NC vs. crystalline for PTI) has a major impact on the structure of the Pt sites. In fact, surprisingly, crystalline supports provide a more heterogenous environment. We want to note that the study was focused on N-doped carbons because they are well-known materials to stabilize single atoms and are probably the largest class of SACs.

However, following the comment of the reviewer, we have decided to extend and show that this methodology can also be extended to oxide materials with so-called 'single-site' catalyst structure;

the data is now shown in Figure 4b and Figure S13 in the revised version of the manuscript and the SI.

Figure S4. Static ^{195}Pt NMR signatures, fitted lineshapes, and $\delta_{\text{iso}}\text{-}\Omega$ -distribution maps of Pt@NC-15 after the second annealing step, batch 1 and 2, Pt@PTI, and Pt@SiO₂.

Figure 4. Reproducibility, alternative supports, and evolution during catalysis via NMR fingerprints. $\delta_{\text{iso}}\text{-}\Omega$ -sampling maps with NMR fingerprints for **a**, two different batches of Pt@NC-15, and **b**, PTI and SiO₂ supports, with illustrations of the structural fragments in the respective top panels. In the $\delta_{\text{iso}}\text{-}\Omega$ -sampling map for Pt@SiO₂, the experimental data point¹³ for a C/O-mixed Pt environment is additionally given for reference. **c**, Monitoring the hydrochlorination of acetylene with Pt@NC-1 as a function of time on stream at t_0 (pristine), t_1 (after 1 h), and t_{12} (after 12h). Productivity vs time in the left panel. $\delta_{\text{iso}}\text{-}\Omega$ -sampling maps with NMR fingerprints for t_0 , t_1 , and t_{12} in the right panels.

We have added the following discussion in the revised version of the manuscript:

Similarly, this methodology can be applied to oxide-supported materials; here illustrating with a single-site catalyst Pt@SiO₂, prepared by grafting (COD)Pt(OSi-(OtBu)₃)₂ on silica. The average CS-tensor parameters ($\langle\delta_{\text{iso}}\rangle/\langle\Omega\rangle = -3900/8500$ ppm), i.e., the center being shifted from the $\Delta\Omega/\Delta\delta_{\text{iso}}=3$ line, document a C/O-mixed chemical environment of the present Pt sites, in agreement with DFT computations (see **Fehler! Verweisquelle konnte nicht gefunden werden.**b) and previous reports,^{13,17} while Q_{ICE} parameters indicate the presence of various local environments as expected for the amorphous silica support (**Figure 4b**, right panel).

3) As well stated in the abstract, such an approach should allow to establish structure/catalytic property relationships. The reader is frustrated not to find any. Moreover, such studies in modern catalysis today require in-situ or operando approaches (due to the dynamics of these systems). I have not found any comment from the authors concerning the applicability of the technique to in-situ conditions (the relatively long times for the analysis constitute a real handicap).

We thank Reviewer #3 for their comment. As discussed above in an answer to Reviewer 2, we have extended our work to monitor how the coordination environment evolves upon catalysis and have done so with a low loading catalyst (1 wt%). The new results are now included in the revised version of Figure 4. Overall, we show that our approach is applicable to ex situ monitoring of catalysts, which is a very important step forward. This methodology is therefore applicable to draw structure-activity relationship and to provide molecular-level understanding of deactivation processes.

These considerations make me believe that this work, although of quality, should be published in a more specialized journal.

We respectfully disagree with the reviewer, and justify above why we consider this work to be of importance beyond the chemistry world. Furthermore, we hope that the added data provided in the current revision spark more enthusiasm in the reviewer.

Referees' comments:

Referee #1 (Remarks to the Author):

The authors have provided comprehensive and detailed responses to all of my raised questions, remarks, and suggestions for improving their work. Moreover, as suggested by Reviewer 2, the now included data of NMR spectroscopic monitoring of the catalytic performance for Pt@NC (1 wt%) in gas-phase hydrochlorination of acetylene (Figure 4c), really highlights the power of the developed approach as now insights about the catalyst deactivation processes can be made! Also, I find that the authors have made their model approach much clearer as suggested by both Reviewer 2 and myself. I congratulate the authors on their excellent work and I strongly recommend that their work should be accepted for publication in Nature.

We again thank Reviewer #1 for the kind words and their appreciation for our work.

Referee #2 (Remarks to the Author):

The authors have provided thoughtful responses to several of the comments and criticisms in the revised manuscript.

1. The addition of the subsections is helpful
2. The authors have connected their analyses of ^{195}Pt NMR spectra of heterogeneous platinum catalysts with reaction data in Figure 4 and in Figure S11 for the hydrochlorination of acetylene. This is a beneficial improvement, which increases the impact of the manuscript.

We thank Reviewer #2 for acknowledging our responses to their initial comments and are grateful for their appreciation for the additional data provided in our revised version of the manuscript.

3. For the high Pt loadings investigated, it seems questionable that Pt atoms would only be present as single atoms. The authors describe their use of wet-impregnation to introduce Pt into an indistinctly described nitrogen-carbon support; impregnation often leads to aggregation of metals during activation treatments, in this case the annealing procedures, and especially for the high metal loadings of 5 wt% and 15 wt% reported. Such high Pt loadings also tend to obscure features in STEM images, which represent tiny regions that are insufficient to assert that the entire sample contains only isolated Pt atoms. Additional chemisorption or spectroscopic evidence should be provided to establish the extent of Pt dispersion to support the claim of single atom catalysts. That said, however, the emphasis on atomically dispersed Pt seems unnecessary, as the authors' analyses appear to be generally useful for assessing complicated inhomogeneous distributions of environments that would also be present if Pt aggregates are present. The authors' methodology would therefore seem to be suitable generally for analyses of inhomogeneous distributions of Pt environments, including in dispersed nanoclusters.

We agree that metal aggregation during thermal activation at high metal contents is often a challenge in the synthesis of single-atom catalysts. However, in our case, platinum is introduced via a coordination-controlled synthetic approach, in which nitrogen sites on the functionalized carbon support are saturated, and excess metal precursor is removed prior to thermal treatment (see Hai et al., Nat. Nanotechnol. **2022**, 17, 174). This procedure has been extensively validated in previous works, including statistical analysis of atomically dispersed metals via automated

detection in HAADF-STEM images (see Rossi et al. Adv. Mater. **2024**, 36, 2307991). We have now applied the same convolutional neural network-based atom detection tool to the present catalysts (see figure below and now in the Extended Data Figure 2), confirming that the vast majority of Pt (~90%) is present as single atoms, while only a minor fraction is potentially present as dimers or trimers. This is likely an overestimate as it is derived based on metal proximity from a 2D projection image and does not account for overlap effects at different heights. This estimate is conservative as the number of clusters is based on atom proximity and is likely overestimated due to projection effects. No clusters greater than two atoms were detected. Contrary to the reviewer's concern, Pt clusters would appear with greater contrast in HAADF-STEM on a carbon support, and would be readily visible to the trained microscopist. Their absence across multiple regions supports our conclusion.

Extended Data Fig. 2. Automated atom detection analysis of HAADF-STEM images of Pt@NC-5 (top) and Pt@NC-15 (bottom) after the second annealing step. (a) and (c) High-resolution STEM micrographs with machine learning detected atoms overlaid (white circles). (b) and (d) Distribution of the nearest-neighbor distance (d_{NN}) between detected metal sites. The estimated upper bounds for multimer formation are 10.6% and 12.4%, defined by atoms separated by <0.22 nm. As STEM provides 2D projections of 3D structures, this value is known to overestimate the actual degree of multimer formation. The analysis was performed using a previously developed open-access tool with a confidence threshold set to 80%.

Concerning additional validation, conventional chemisorption techniques are not applicable here due to weak binding of probe molecules (e.g. CO) to cationic Pt centers, and the carbon supports are not suitable for IR analysis. However, the XAS and solid-state ^{195}Pt NMR results evidence the predominance of isolated square-planar Pt(II) species. No features indicative of metallic Pt clusters are observed (see e.g., van der Putten et al. from 1992 and Hanna et al. from 2013). That said, we appreciate the Reviewer's observation that our analytical approach could, in principle, be extended beyond single-atom systems, and we have now highlighted this potential as an outlook. However, we focus here on single-atom catalysts, as robust and reproducible synthetic methods for more complex architectures, such as dual atoms or nanoclusters, remain limited.

We therefore prefer not to overstate the method's generality at this stage, but agree that it holds promise for broader application as the field advances.

4. The brief mention of Figure 1 on page 3 is inadequate. The extensive information content of Figure 1 needs to be discussed or the figure should be condensed or omitted. Figure 1b: The caption is incomplete, with no mention or description of the N or Cl tensor schematics or explanation of what the associated colors correspond to Figure 1c: The caption is incomplete with no description of what material the spectrum (left) corresponds to, 2D axis labels are missing (middle), and insufficient description of the significance of the different schematic diagrams (right, presumably sq. planar complexes with different ligands, grey is unclear) or their connection to the middle black square, which seems out of place.

Following the reviewer's and the editors' suggestion, we have revised Figure 1. We have removed 1a and redesigned previous Figure 1b/c in order to better explain the approach. To this end, in a (former b) we have removed one of the two CS tensors and the corresponding ^{195}Pt NMR spectrum, to now include an orbital energy diagram, showing the Pt lone pair and the σ^* orbital. We have included this to more clearly establish the connection between the electronic structure and the CS tensor, and furthermore to emphasize, that for square-planar Pt(II), the energy difference ΔE between these two orbitals modulates the magnitude of the δ_{11} component of the CS tensor. Therefore, it is primarily d_{11} which drives the chemical shift and span, which in turn is the essence of the model we introduce to describe SACs. In b (former c), we have now replaced the nine Pt environments by the distribution of environments formerly included in the original 1a. Here we have also included a color gradient consistent with the colors used in Figure 2. The EM image and the distribution in b is now meant to parallel the energy diagram and the single CS tensor in a. Additionally, we have included in the ^{195}Pt NMR signature of a SAC (lower panel in b) individual ^{195}Pt NMR powder patterns (likewise in color code) to emphasize the superposition of various local Pt environments. Furthermore, we indicate the variation of δ_{11} with a color-coded double arrow to make clear distinction to the spectrum of well-defined Pt site (lower panel a), with clearly-defined δ_{11} position.

5. Figure 2: The authors' response to the repetitive figure panels in Figure 2, while true, does not overcome the fact that the multiple similar spectra are insufficiently discussed and tedious for the reader to have figure out on one's own. Given the terse discussion, little is gained by including so many of them. I recommend that representative examples be included and discussed, referring to the others placed in the Supporting Information. The authors should clarify how the values for are determined: they are not placed at the foot of their respective "blurred" lineshape where it reaches the baseline, but rather appear to be placed relatively arbitrarily where intensity is non-zero. Describe "annealing" conditions in the figure caption at least briefly: in N₂ at 200 oC and then 550 oC.6. Page 6: More detailed description is needed of the NC support: e.g., N content (from XPS data in Table S2) and porosity.

We appreciate the reviewer's perspective here. However, showing these spectra for the different catalysts for the different annealing steps is important to demonstrate the overall change across them. Indication of the average principal axis components show the similarity to the molecular compounds, and the figure has been designed to help the reader to understand the overall concept and guide them through the discussion. Furthermore, we reiterate that these are the only NMR spectra that we show, as we in the following only use the 2D maps, into which the NMR spectra have been converted (via Monte Carlo). We want to bring to the attention of the reviewer

again to the fact that the position of the singularities is the result of the best numerical fit obtained by the fitting algorithm. This is briefly described in the main text, and exhaustively discussed in the Supporting Information.

The annealing conditions are detailed in the Methods section. To maintain an acceptable length of the figure caption, we now in the revised version mention the temperatures in the main text.

Lastly, we have extended the description of the nitrogen-doped carbon (NC) support, including the surface area determined by N₂ physisorption and the surface N content derived from XPS. This is now included in the Supporting Information in Section 1.2 and Table S1 and S4.

7. Page 7, bottom: What is the nature of Pt bonding with the N atoms in the carbon support? Schematic diagrams of the local bonding structures that are indicated by the data and the changes that occur with annealing would be helpful to include.

As discussed, the CS-tensor parameters provide information of the average local environments of Pt sites. Considering that we show that SACs are always a distribution of sites, we do not think it is possible to provide a representation for specific structures. The local structure of these catalysts is more comprehensively described by the 2D maps, and it is why we have translated the wide-line NMR signatures (from Figure 2) into these 2D representations. We find the 2D maps to visualize well the development of the local structures and the heterogeneity.

8. Figure 3: Explain why the 2nd annealing treatment narrows the heterogeneous distribution(s) in Fig. 3d, whereas annealing leads to broader features in Fig. 3f. Could the latter be due to Pt aggregation into clusters? If not, how can that be ruled out?

We thank Reviewer #2 this question. This is of course an important observation, and we have discussed this in our manuscript, already at the initial submission:

“For Pt@NC-15, the increase of both σ (from 400 ppm to 700 ppm) and ρ (from 0.7 to 0.9) point in this case to the almost full removal of Cl in the Pt(II) environment already after the first annealing step and to the integration of Pt within the support upon the second annealing step. Anchoring is expected to yield a larger variety of coordination environments, induced by the intrinsic support inhomogeneity.”

We think that this sentence provides a possible explanation for these observations.

9. Figure 4: Reproducibility is necessary and presumed. The new results don't belong in the main text; Fig. 4a and its short discussion are more appropriate for the Supporting Information. It is not clear what atoms Pt, N, C, O, H, Cl, etc. are represented in the 3 structures; all types of atoms need to be explicitly defined, especially those associated with Pt and its ligands.

Note that the reproducibility panel was already provided in the first submission and that the term reproducibility refers here to evaluating the reproducibility of the synthesis across two materials prepared based on their NMR signatures (and not to recording two spectra of the same sample). Therefore, we think it is an important message and we decided to keep this data in the figure and the main text because it illustrates the capacity of this NMR methodology to control the reproducibility and the quality of different batches in terms of local Pt sites.

Following the reviewer's suggestion, we have revised the figure legend to include an explicit assignment of the shown atom types regarding the color code.

10. Page 10: The catalytic reaction results are an important addition to the manuscript. The connection to catalyst deactivation is interesting.

We thank Reviewer #2 for their appreciation of the new data.

11. The claim that the distribution of parameters is a “fingerprint” of molecular composition is an overstatement. Whereas fingerprints are unique to a given individual, this cannot be said (or at least has not convincingly been shown) for the combination of broad features that the authors’ analytical approach determines for different catalysts. Based on the discussion provided, the uniqueness of the values obtained from such overlapping spectral features seems doubtful. This does not diminish the importance of the quantitative insights obtained on these challenging platinum catalysts, but use of “fingerprint” seems exaggerated and does not seem justified.

Following the reviewer’s comment, we have revised our terminology, and consistently refer to NMR spectra or (powder) patterns when discussing NMR spectra of molecular compounds with well-defined metal sites, and to NMR signatures when discussing the NMR spectra of supported Pt species because of the inhomogeneous distributions of Pt sites (formally phrased fingerprints). We have also revised the title of the manuscript accordingly to “**Coordination environments of platinum single atom catalysts from NMR signatures**”.

Overall, the strengths of the manuscript remain notable: its combined Monte Carlo, DFT and NMR analyses provide important quantitative insights on the local structures and bonding environments that are correlated with the preparation chemistry and catalytic reaction properties of Pt supported on nitrogen-doped carbons. The manuscript is improved, but the authors should consider the above comments before publication in Nature.

We thank Reviewer #2 for acknowledging the quality and impact of our work.

Referee #3 (Remarks to the Author):

I appreciate the effort made by the authors to extend the scope of supports, at least to one oxide (SiO₂) and especially to introduce chemical reactivity results even if in ex situ conditions (hydrochlorination of acetylene). I think that these important additions can contribute to a better impact of their work.

We thank Reviewer #3 for their appreciation of the additional data provided in our revised version of the manuscript.